# Transcription recycling assays identify PAF1 as a driver for RNA Pol II recycling

Zhong Chen [1✉], William Hankey [1], Yue Zhao [1,2], Jeff Groth[1], Furong Huang[1], Hongyan Wang[1], Alexandre Rosa Campos[3], Jiaoti Huang[1], Robert G. Roeder[4] & Qianben Wang [1✉]

RNA Polymerase II (Pol II) transcriptional recycling is a mechanism for which the required factors and contributions to overall gene expression levels are poorly understood. We describe an in vitro methodology facilitating unbiased identification of putative RNA Pol II transcriptional recycling factors and quantitative measurement of transcriptional output from recycled transcriptional components. Proof-of-principle experiments identified PAF1 complex components among recycling factors and detected defective transcriptional output from Pol II recycling following PAF1 depletion. Dynamic ChIP-seq confirmed PAF1 silencing triggered defective Pol II recycling in human cells. Prostate tumors exhibited enhanced transcriptional recycling, which was attenuated by antibody-based PAF1 depletion. These findings identify Pol II recycling as a potential target in cancer and demonstrate the applicability of in vitro and cellular transcription assays to characterize Pol II recycling in other disease states.

[1] Department of Pathology and Duke Cancer Institute, Duke University School of Medicine, Durham, NC 27710, USA. [2] Department of Pathology, College of Basic Medical Sciences and First Affiliated Hospital, China Medical University, Shenyang 110122, China. [3] Sanford Burnham Prebys Medical Discovery Institute, La Jolla, CA 92037, USA. [4] Laboratory of Biochemistry and Molecular Biology, The Rockefeller University, New York, NY 10065, USA. ✉email: zhong.chen128@duke.edu; qianben.wang@duke.edu

Eukaryotic transcription progresses from initiation to the elongation and termination phases. After the initial transcription cycle, RNA polymerases (I, II, and III) repeatedly transcribe the same gene and generate multiple RNA copies from the DNA template, contributing to robust overall transcriptional output[1,2]. This transcription recycling process is best exemplified by RNA polymerase III (Pol III) recycling, which produces a large number of tRNA and 5S rRNA transcripts required for translation. Using in vitro and cell-based approaches, numerous factors have been identified as regulators of Pol III recycling, including TFIIIB, TFIIIC, Maf1, and La[2–4]. Compared with Pol III recycling, RNA polymerase II (Pol II) recycling is an overlooked yet key transcription process that significantly affects Pol II mRNA output. A major obstacle for studying Pol II recycling has been the lack of an appropriate in vitro model system. Currently, in vitro Pol II transcription systems are mainly used to study promoter-bound, retained, and reutilized protein factors driving Pol II reinitiation, the first step of Pol II recycling, after the initial Pol II transcription from the template[5]. For example, an in vitro study using an immobilized promoter assay found that pre-initiation complex (PIC) lacking Pol II and TFIIF remains on the promoter after initial transcription, forming a scaffold to facilitate the assembly of the reinitiation complex[6]. Further studies using a defined in vitro transcription system showed that the Mediator CDK8 subcomplex inhibits reinitiation of Pol II transcription from the PIC scaffold complex[7]. Despite their usefulness in the study of reinitiation, a common limitation of the existing in vitro Pol II transcription systems is that they are unable to discover proteins possibly regulating Pol II from reinitiation to travel throughout the gene and handing back to the promoter (i.e., Pol II recycling).

To identify putative proteins regulating Pol II recycling, we developed an in vitro transcription system using immobilized DNA templates with human cell nuclear extracts to isolate proteins participating in different transcription processes, including PIC formation, initial transcription, multi-round transcription, and recycling. Notably, while previous studies use inhibitory molecules (e.g., the detergent sarkosyl) or devised templates to separate transcription initiation from reinitiation[5], we utilize a distinct non-devised two-template approach to separate multi-round transcription on the same DNA template from recycling onto the new DNA template without using detergents to block the transcription processes. Interestingly, among those proteins recycled with Pol II onto the new template is the human polymerase-associated factor 1 complex (PAF1C). PAF1C has been traditionally been studied as a group of proteins regulating proximal Pol II pausing as well as transcription elongation and termination[8]. Importantly, our in vitro discovery of a novel functional role of PAF1C in driving Pol II recycling on human genes is complemented by a strategy tracking PAF1C-faciliated Pol II progression in real-time in human cells. Significantly, PAF1 is elevated in prostate tumors and required for the higher transcription recycling detected in prostate tumors compared with adjacent non-tumor prostate tissues. Altogether, the in vitro- and cell-based transcription recycling system we have designed opens the door for researchers to study an underappreciated but important area of transcriptional regulation: Pol II transcription recycling.

## Results

**Development of an in vitro transcription recycling assay.** We developed a two-template transcription recycling assay to study Pol II recycling in vitro. The system was designed to facilitate the separation of Pol II transcription into pre-initiation, initial transcription, and multi-round transcription phases (all taking place on the first DNA template) as well as a recycling phase (taking place on the second DNA template) (Fig. 1a). This contrasted with the previous two-template in vitro transcription assays designed to study the role of basal transcription factors that facilitate transcription reinitiation while predominantly remaining associated with the first template[9,10]. Of note, specific initiation was observed downstream of the promoters in both templates (Supplementary Fig. 1a–d). In our two-template recycling assay, we first incubated biotinylated DNA Template 1 with HeLa nuclear extracts. The preinitiation complex (PIC) was formed by incubating bead-immobilized template DNA with nuclear proteins, while the subsequent addition of nucleoside triphosphate (NTP) mix was required to trigger progression into the initial transcription phase. Following a 60 min period of multi-round transcription, Template 1 was washed and a new transcription buffer was added back to resume transcription. Then, DNA-associated factors were isolated in three sequential elutions, and these transcription intermediates were incubated immediately with a second DNA template (Template 2) (Supplementary Fig. 1e). Of note, we did not add any new transcription components (i.e., nuclear extract, proteins) to Template 2. This experimental design allowed us to study whether transcription proteins from Template 1 (i.e., run-off proteins from the Template 1 gene body and drop-off proteins from the Template 1 promoter) can be "recycled" onto Template 2 and drive transcription from this second template. As expected, unphosphorylated Pol II was depleted and Ser 2 phosphorylated Pol II was enriched in the recycling phase (Fig. 1b). Interestingly, RT-PCR analysis found that the first and second elutions that captured factors from Template 1 could both efficiently drive transcription from Template 2, indicating robust Pol II transcription recycling (Fig. 1c). The failure of the third elution to continuously drive Template 2 transcription suggested that Pol II recycling is not an unlimited transcription process (Fig. 1c). As a control, transcription products from Template 1 were also detected in each elution, but did not increase in abundance during the 60 min recycling incubation with Template 2, confirming the absence of Template 1 from the recycling samples (Supplementary Fig. 1f). Together, these findings demonstrated that Pol II can be recycled in vitro and that the specific Pol II recycling process can be isolated and studied using our in vitro two-template recycling system.

**Proteomic identification of factors associated with RNA Pol II transcriptional recycling in vitro.** One advantage of our in vitro transcription recycling system is that the immobilized template design enables isolation and identification of protein factors associated with the DNA at each phase using mass spectrometry. Template-associated proteins from each of the four phases (Fig. 1a) were resolved using two-dimensional gels and identified by LC-MS-MS/MS in biological triplicates with technical triplicates for each phase. Unsupervised clustering analysis of these samples was applied to identify differentially enriched proteins for each phase (Supplementary Fig. 2a–d). Of note, sample swap analysis (purposely matching up samples from different conditions as "replicates") failed to reproduce the results shown in Supplementary Fig. 2d (Supplementary Fig. 2e, f), demonstrating that the differentially enriched proteins are convincingly and reproducibly identified and not found by chance. To confirm the validity of our proteomic approach, we first performed a partial procedure with Template 1 only, focusing on the proteins associated with the PIC phase and the multi-round transcription phase (Supplementary Fig. 3a–c). Proteins enriched (>2-fold) in the PIC phase included components of the TFIID complex such as

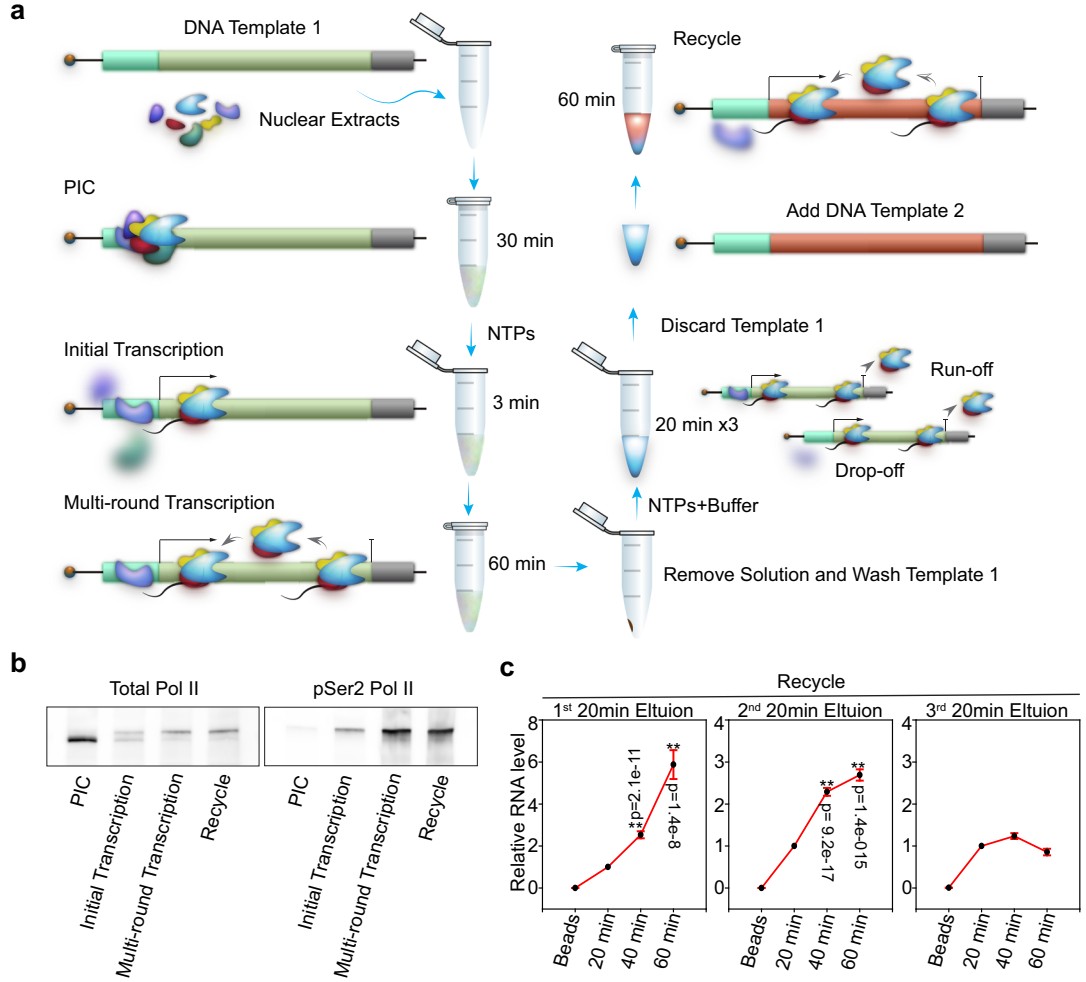

**Fig. 1 In vitro transcription recycling system. a** Schematic of the in vitro transcription recycling assay. **b** Western blot analysis of total Pol II and phosphorylated Pol II bound to the template in individual transcription steps. These images are representative of two independent experiments. **c** In vitro transcribed RNA products from the 2nd template during the Recycle step were analyzed by reverse transcription followed by qPCR quantification. Values were expressed as the mean ± SEM of three independent experiments. *p*-values were calculated using two-tailed Student's *t*-test comparing the 20 min group to other experimental groups, **\**p* < 0.01. Source data are provided as a Source data file.

the TATA-Box binding protein (TBP), TBP-associated factors (TAF) 1–10, and other general transcription factors and the Mediator complex required for RNA Pol II activity, such as GTF2F1/2 and MED6[11,12] (Supplementary Fig. 3b, c). Among the proteins enriched in the multi-round transcription phase was SPT6H, a protein traveling with elongation Pol II[13,14] (Supplementary Fig. 3b, c). The enhanced binding of TOP2A and TOP2B was consistent with the requirement for topoi-somerases at the elongation stage of transcription[15]. Also enriched was the termination factor XRN2, which dissociates the elongating Pol II complex from the gene during the ter-mination process[16] (Supplementary Fig. 3b, c). These data indicated that the immobilized template experimental design is very useful for discovering proteins enriched within a par-ticular transcription process, providing a basis for identifying proteins involved in transcription recycling.

Having established the two-template in vitro transcription recycling system and demonstrated that the proteins associated with the immobilized templates can be isolated and identified by mass spectrometry, we next performed a full experimental procedure to characterize proteins on both Template 1 and Template 2. This resulted in the isolation of proteins associated with Template 1 during the PIC and multi-round transcription

phases, and with Template 2 during the recycling phase (Fig. 2a). LC-MS-MS/MS was used to identify proteins exhibiting two-fold enrichment in or depletion from the recycling phase relative to the PIC or multi-round transcription phase (Fig. 2b). Among the enriched recycling factors enriched on Template 2 are the polymerase associated factor (PAF1) complex components PAF1, LEO1, CTR9, and CDC73[8]. To alleviate concerns that these proteomics results might be template- or promoter-specific, the experiment was reproduced using an alternative template (Supplementary Fig. 4a), yielding closely related protein hits including the PAF1 complex and similarities in fold-changes for each hit from the two experiments (Supplementary Fig. 4b–d). While existing methods have identified roles of the PAF1 complex in regulating Pol II pausing[17,18], elongation[19–21], cleavage, and polyadenylation[22–24], our novel approach has uncovered a novel function of the PAF1 complex as a putative regulator of Pol II recycling.

**PAF1 travels dynamically through the gene body during transcription in human cells.** While the in vitro transcription recycling assay coupled with mass spectrometry successfully identified proteins involved in Pol II recycling, this approach was not able to distinguish whether individual proteins found on

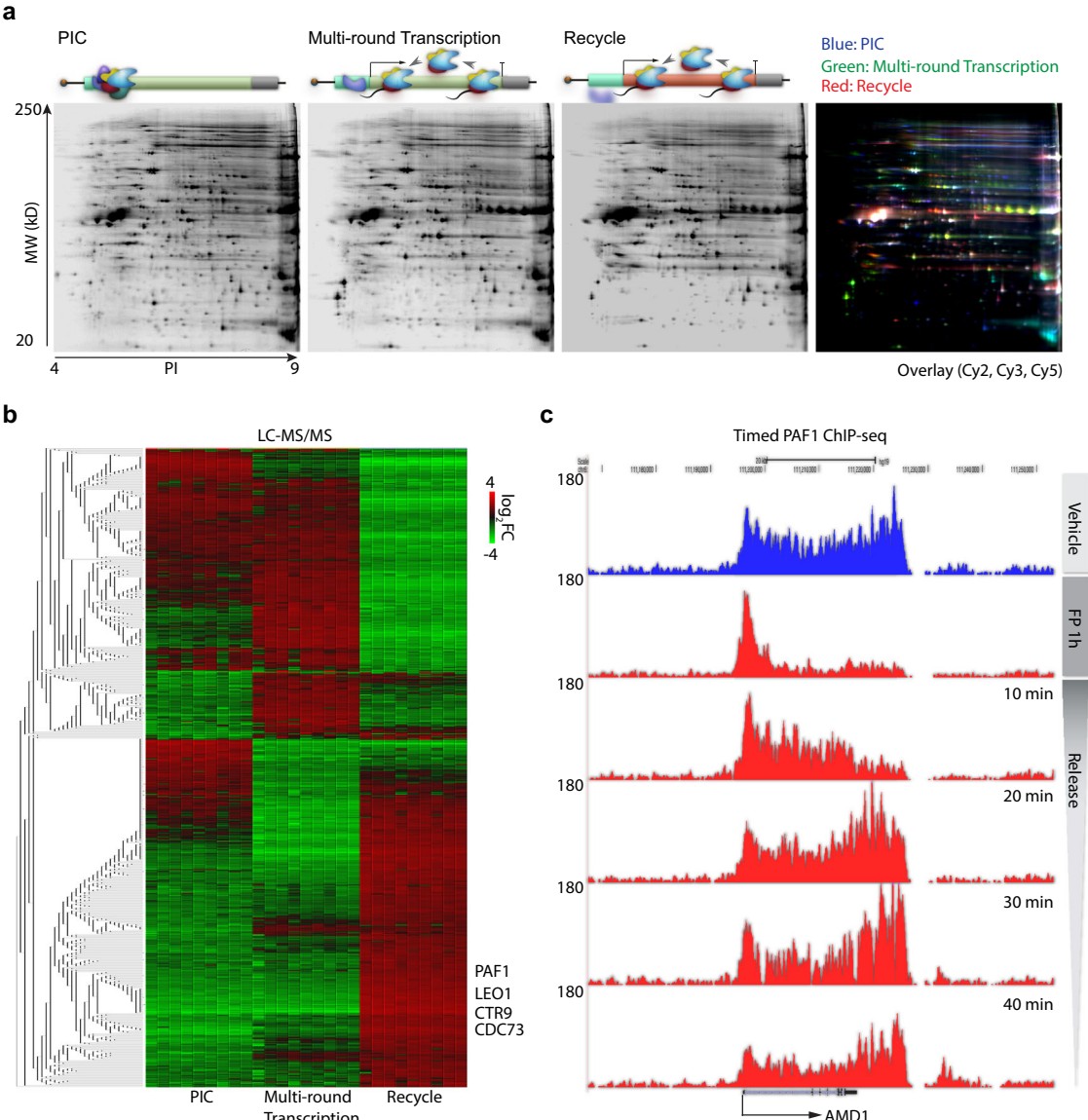

**Fig. 2 Profiling of in vitro transcription on the sequential template. a** Representative two-dimensional gels showing proteins bound to the sequential template in individual transcription steps. **b** Heatmap showing profiling of proteins identified by LC-MS-MS/MS across all individual transcription steps on the sequential template. **c** UCSC Genome Browser views of a representative PAF1 time course ChIP-seq in LNCaP-abl cells. Source data are provided as a Source data file.

Template 2 had left Template 1 by a drop-off mechanism from the promoter or by a run-off mechanism following travel along the gene body. To ask whether the PAF1 complex travels throughout the gene and recycles back, we performed a ChIP-seq time course for the core subunit PAF1 in the prostate cancer cell line LNCaP-abl, and monitored PAF1 distribution throughout loci before and after flavopiridol (FP) inhibition and at various time points following release from the drug. FP inhibits CDK9 phosphorylation of RNA Pol II to block the ability of the PIC to transition to the elongation phase of transcription[25]. Analysis of the representative *AMD1* locus showed a population of PAF1 trapped at the promoter following 1 h FP treatment, and traveling through the gene body over a series of time points following release (Fig. 2c). PAF1 occupancy over 5000 human RefSeq genes in LNCaP-abl cells (Supplementary Fig. 4e) showed enrichment at transcription start sites (TSS) and absence from the gene body following 1 h FP treatment. Since FP treatment compromises new PAF1 recruitment, new Pol II phosphorylation and Pol II escape

from the paused state without affecting existing Ser2 phosphorylated Pol II (associated with PAF1) traveling within the gene body[13,17,26], our timed PAF-1 ChIP-seq results suggest that PAF1 may travel and recycle back to the TSS. 10 and 40 min recovery following FP removal progressively restored gene body occupancy and overall PAF1 distribution to match the vehicle treatment condition. Release from FP also increasingly restored the balance between PAF1 accumulating near the TSS and PAF1 in the remainder of the gene, expressed as a Pausing Ratio (Supplementary Fig. 4f). These data indicate that the focused dynamic ChIP-seq analysis nicely complements the unbiased in vitro transcription recycling coupled with proteomics analysis. Importantly, these in vitro and cell-based data together indicate that PAF1 may actively participate in transcription recycling. In addition, given that PAF1 and LEO1 are structurally tightly associated[27] and that PAF1 and LEO1 co-bound tightly through transcribed genes (Supplementary Fig. 4g–i), PAF1 and LEO1 may collaborate during the process of transcription recycling.

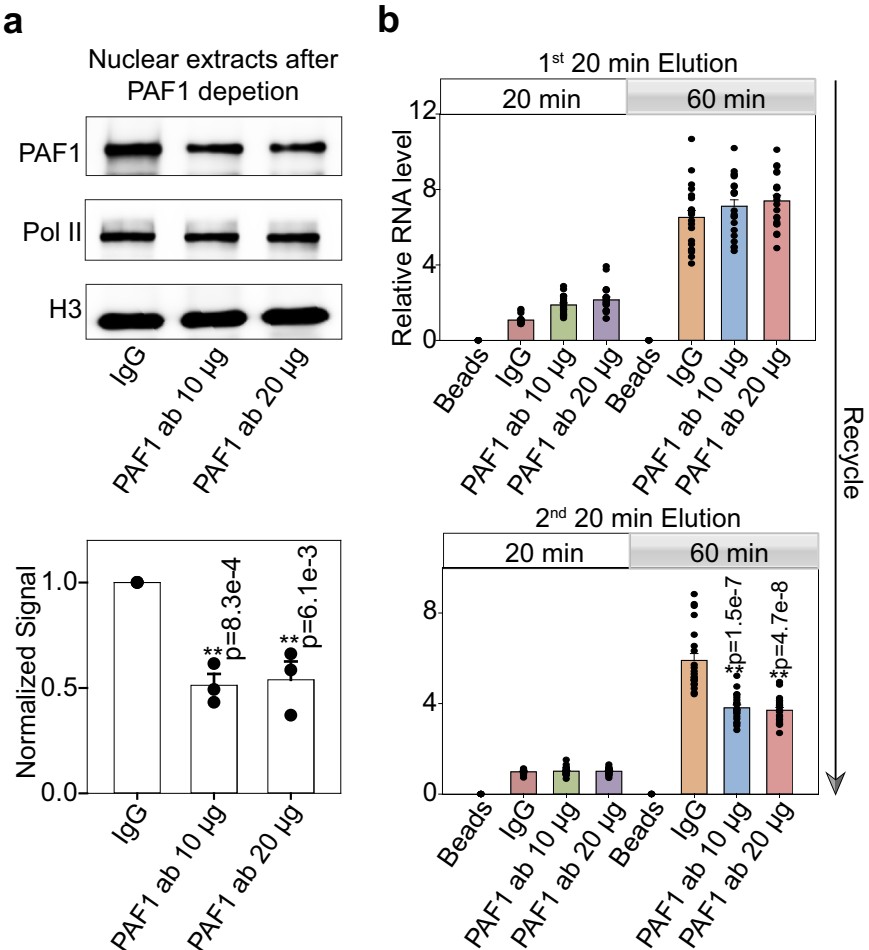

**Fig. 3 PAF1 depletion attenuates transcriptional recycling in vitro. a** Nuclear extracts from LNCaP-abl cells were incubated with PAF1 antibodies to moderately deplete PAF1 protein, and then were analyzed by western blot. Histone H3 was used as an endogenous loading control. The graph shows the quantification of PAF1 expression. Values are expressed as the mean ± SEM of three independent experiments. **b** The transcription recycling assay was performed using nuclear extracts with or without PAF1 depletion. In vitro transcribed RNA products from the 2nd template during the Recycle step were analyzed by reverse transcription followed by qPCR quantification. Values were expressed as the mean ± SEM of three independent experiments. *p*-values were calculated using two-tailed Student's *t*-test, **p < 0.01. Source data are provided as a Source data file.

**PAF1 depletion attenuates RNA Pol II transcriptional recycling in vitro**. To investigate whether PAF1 plays a causal role in Pol II recycling, we examined the effect of PAF1 depletion on transcription recycling in vitro. Moderate depletion of PAF1 (Fig. 3a and Supplementary Fig. 5a) was achieved by incubation of 10 or 20 μg of PAF1 antibody (or whole IgG control) with LNCaP-abl nuclear extracts. Western blotting with an alternative PAF1 antibody confirmed depletion, while total RNA Pol II levels were unchanged (Fig. 3a and Supplementary Fig. 5a). The resulting nuclear extracts were subjected to the transcriptional recycling assay (Fig. 1a) to evaluate the ability of their first and second elutions from Template 1 to transcribe Template 2 (Fig. 3b). Production of RNA by the proteins contained in the first elution did not change significantly with PAF1 depletion, suggesting that PAF1 and some highly enriched factors in the first elution play redundant roles in the regulation of recycling on naked DNA templates. The second elution, however, exhibited a significant decrease in transcriptional activity from the IgG control to each antibody-depleted condition, indicating that PAF1 removal significantly attenuates transcriptional recycling. On the other hand, PAF1 depletion did not significantly attenuate transcriptional activity on Template 1 (Supplementary Fig. 5b). We also performed the transcriptional recycling assay using

nuclear extract of non-cancerous BPH1, a human benign prostatic hyperplasia cell line that expressed PAF1 at a lower level than LNCaP-abl (Supplementary Fig. 5c). While transcriptional activity from the second elution was more than three-fold lower in transcription recycling assays using nuclear extracts from BPH1 rather than LNCaP-abl, moderate depletion of PAF1 from BPH1 nuclear extract still significantly decreased transcriptional recycling, although to a lesser extent (Supplementary Fig. 5d). Interestingly, co-depletion of the LEO1 and WDR61 subunits of the PAF1 complex did not significantly enhance attenuation of recycling by PAF1 depletion (Supplementary Fig. 5e). Since depletion of PAF1 or LEO1 but not WDR61 caused depletion of other PAF complex subunits and led to decreased transcription recycling (Supplementary Fig. 5f, g), our results indicate that a PAF1- and LEO1-containing PAF1 complex is important to enable Pol II transcription to recycle, while WDR61 is dispensable for transcription recycling.

**PAF1 is required for RNA Pol II transcriptional recycling in human cells**. We next examined the relevance of PAF1 to cellular RNA Pol II transcriptional recycling. We used two complementary timed Pol II ChIP-seq strategies[13] to study the role of PAF1 in Pol II traveling and recycling dynamics. Briefly, since FP

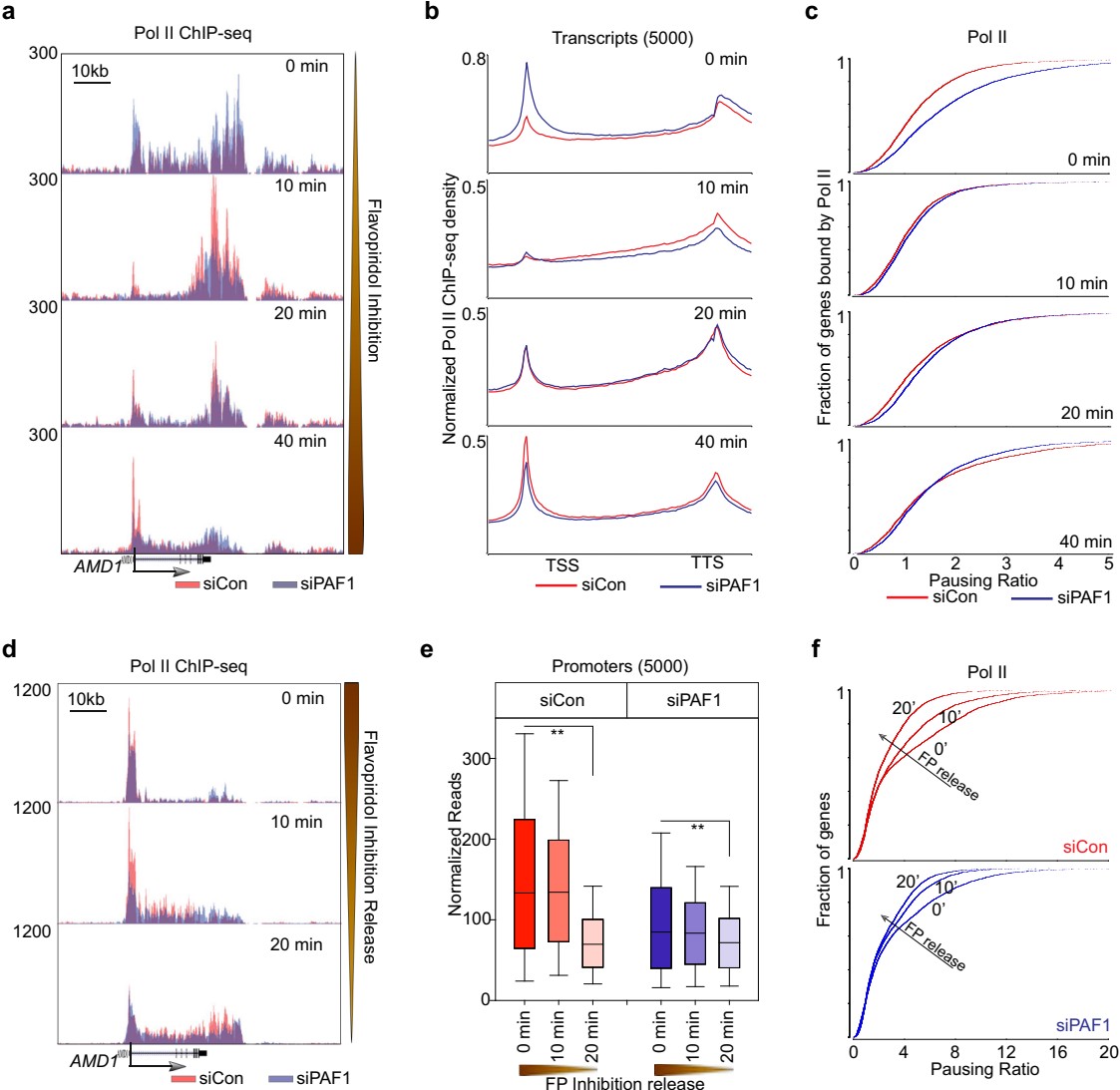

**Fig. 4 PAF1 is required for Pol II transcriptional recycling in human cells. a** Genome Browser views illustrating Pol II (Ser2) ChIP-seq signal changes in LNCaP-abl cells treated with siControl or siPAF1 during Flavopiridol inhibition. **b** Average Pol II (Ser2) ChIP-seq signal densities over the scaled 5000 human RefSeq genes in LNCaP-abl cells. **c** Pausing ratio of Pol II (Ser2) was calculated, sorted, and plotted in LNCaP-abl cells with timed FP inhibition. **d** Genome Browser views illustrating Pol II (Ser2) ChIP-seq signal changes in LNCaP-abl cells treated with siControl or siPAF1 during release from Flavopiridol inhibition. **e** Box plots show Pol II (Ser2) ChIP-seq signal densities in 5000 promoter regions during timed FP inhibition release. Box plots show median (center), 25th/75th percentiles (box bounds), 10th/90th percentiles (whiskers). The values beyond the whiskers are not shown in the boxplot. *p*-values were calculated using one-way ANOVA on ranks, **$p < 0.001$. **f** Pausing ratio of Pol II (Ser2) was calculated, sorted and plotted in LNCaP-abl cells during timed FP inhibition release. Source data are provided as a Source data file.

treatment inhibits the occurrence of new Pol II phosphorylation events but has no effect on existing Ser2 phosphorylated Pol II traveling within gene bodies[26], FP can be used to block the release of new Pol II into productive elongation in a timed manner. In this way, the "retreating wave" of Ser2 phosphorylated Pol II already present in the gene bodies can be detected by timed Pol II (Ser2) ChIP-seq to study Pol II traveling dynamics. Another way to study Pol II traveling is to first treat the cells with FP for 1 h. After wash-out of FP, the "emerging wave" of newly released Ser 2 phosphorylated Pol II can be studied by timed Pol II ChIP-seq to track Pol II traveling dynamics[13]. In our studies, Ser 2 phosphorylated Pol II ChIP-seq time course was generated during 0-, 10-, 20-, or 40 min FP treatment to block the release of new Pol II into gene bodies (Fig. 4a–c and Supplementary Fig. 6a–d) and during 0-, 10-, 20- or 40 min recovery time points after wash-out of FP to release Pol II into the gene bodies (Fig. 4d–f and

Supplementary Fig. 6e, f) in LNCaP-abl cells transfected with either non-targeting control siRNA or low-concentrations of siPAF1 that did not affect cell viability (Supplementary Fig. 6a). During the 40-minute FP treatment, the Ser2 phosphorylated Pol II signal from the siControl condition (red) progressed through the gene body and simultaneously accumulated at the TSS of the representative *AMD1* locus, while the siPAF1 condition (blue) showed marked reduction of Ser2 phosphorylated Pol II recycling to the TSS (Fig. 4a). Scaled analysis of 5000 human RefSeq genes revealed a similar pattern of reduced Ser2 phosphorylated Pol II recycling in siPAF1-transfected cells over the time course of FP treatment (Fig. 4b). Pausing Ratios further indicated that PAF1-silencing led to a decreasing proportion of promoter-associated RNA Pol II as FP treatment progressed (Fig. 4c), indicating that less Ser2 phosphorylated Pol II recycled back to the promoter in siPAF1-transfected cells than in control cells. Consistent with

attenuated recycling in PAF1 silenced cells, ChIP-seq reads during drug washout time points indicated a reduced release of RNA Pol II into the gene body (Fig. 4d). ChIP-seq signal across 5000 human RefSeq genes during FP treatment similarly showed attenuation of RNA Pol II accumulation in promoter regions between the 10 min time point and the 20- and 40 min time points (Supplementary Fig. 6b, c) and reduced movement out of those promoter regions after release from FP (Ser2 phosphorylated Pol II signal in the promoter regions after 20 min of release from FP decreased by 47.7 and 15.5% in the siControl group and the siPAF1 group, respectively) (Fig. 4e). Pausing Ratios showing the proportion of TSS-associated RNA Pol II during FP inhibition time points (Supplementary Fig. 6d) and following FP removal (Fig. 4f) further indicated that PAF1 silencing attenuated promoter accumulation of Ser2 phosphorylated RNA Pol II during FP treatment. Average RNA Pol II signal density at TSS and transcription termination sites (TTS) following release from FP inhibition supported the conclusion that PAF1 silencing prevents Ser2 phosphorylated Pol II travel from TSS to TTS (Supplementary Fig. 6e, f). Together, our dynamic Pol II ChIP-seq analyses after timed FP inhibition or timed FP wash-out suggest that PAF1 is required for cellular Pol II recycling.

**PAF1 contributes to elevated RNA Pol II recycling in tumor samples.** To test the clinical relevance of the PAF1 complex to RNA Pol II transcriptional recycling in the context of cancer, immunohistochemistry (IHC) was first performed on tissue microarrays to detect PAF1 in normal prostate ($n = 32$), histologically normal tissue adjacent to a tumor (NAT, $n = 16$) or tumor tissue ($n = 322$). Representative IHC staining of each tissue type is shown (Fig. 5a). Interestingly, IHC analysis revealed significant over-expression of PAF1 in prostate tumor samples relative to both NAT and normal prostate tissue controls (Fig. 5b). To directly examine the importance of the PAF1 complex to RNA Pol II transcriptional recycling in patient tissues, five human primary prostate cancer tissues and paired histologically normal tissues adjacent to the tumor (NAT) were used in an in vitro transcription recycling assay. Nuclear extracts were prepared from pooled prostate cancer vs. non-tumor adjacent tissues, then incubated with whole IgG control or PAF1 antibody, which resulted in moderate PAF1 depletion (Fig. 5c and Supplementary Fig. 7a). Nuclear extracts were then subjected to the transcription recycling assay, measuring the transcriptional output from Template 1 (multi-round transcription) and Template 2 (recycling) using reverse transcription and qPCR. Nuclear extracts from tumor tissue exhibited significantly higher transcriptional output from the recycling step (Fig. 5d) at both the 20 min time point and especially the 60 min time point. This was dramatically attenuated by PAF1 depletion, which returned recycling-dependent transcription to near-normal levels (Fig. 5d). Transcriptional output from the multi-round transcription step (Supplementary Fig. 7b) was also significantly higher from the prostate cancer tissues relative to the adjacent non-tumor tissue controls. In conclusion, prostate tumors exhibit higher rates of RNA Pol II transcriptional recycling relative to paired NAT, and this effect is at least partially PAF1-dependent.

## Discussion

Published in vitro reconstituted RNA Pol III transcriptional recycling assays have proven the importance of factors such as TFIIIB[28] and TFIIIC[29] to Pol III transcriptional recycling and gene transcription rates. Pol III is particularly applicable and amenable to transcriptional recycling studies as a dedicated polymerase for short, highly transcribed targets such as tRNAs and the 5S rRNA[1]. In contrast, RNA Pol II recycling has

remained poorly characterized, in part due to the absence of a corresponding in vitro system capable of identifying putative recycling protein components. This study reports the development of a system reconstituting RNA Pol II-based transcription from nuclear extracts, using sequential, immobilized DNA templates to separate the process into PIC, initial transcription, multi-round transcription, and recycling phases. This strategy enables template-associated factors to be isolated and identified at each step in an unbiased, proteomics-based manner.

The current methodology incorporates advantageous features from previous in vitro assays of RNA Pol II transcription, such as the use of cell extracts (nuclear in this case) rather than purified proteins as an unbiased source of transcriptional factors[30], immobilization of biotinylated templates on streptavidin-coated beads[31], and use of dual templates encoding distinct transcripts[9,10]. A novel feature of this assay is the use of those distinguishable templates to separate the multi-round transcription and recycling phases (Fig. 1a) in order to facilitate the identification of recycling factors while eliminating the challenge of separating and distinguishing transcriptional steps on a single template using common in vitro elements such as G-less cassettes[32] or detergent-based blocking of RNA Pol II initiation/reinitiation[33]. A sequential two-template design in which transcription intermediates are transferred from the first template to the second template without adding any new transcription components provides a useful tool to study recycling.

As a proof of principle, the system has been used to proteomically identify components of the PAF1 complex among factors involved in RNA Pol II recycling (Fig. 2 and Supplementary Fig. 4). This finding establishes a link between the PAF1 complex and RNA Pol II recycling, and makes a compelling case for the utility of this methodology for future studies seeking to identify novel proteins functioning in a particular phase of RNA Pol II transcription. One limitation of the methodology is that the synthetic templates from this in vitro experimental design are not fully chromatinized, so that identified factors require validation in actual chromatin. Another limitation is the inability to distinguish which proteins isolated from Template 2 are truly run-off factors that have accompanied RNA Pol II through the gene body of Template 1 and facilitated its recycling, and which proteins simply drop off of and drop onto templates dynamically to participate in transcription. Nonetheless, coupling this in vitro transcription recycling system with mass spectrometry provides an unbiased and powerful means to discover putative factors involved in transcription recycling for validation in cells. In the case of newly identified proteins such as PAF1, cellular transcription assays are required to complement the in vitro studies and to distinguish between drop-off and run-off factors. Timed ChIP-seq at multiple time points during and after FP treatment have established PAF1 as a run-off factor that travels through the gene body with Ser 2 phosphorylated Pol II (Fig. 2c and Supplementary Fig. 4e, f). Antibody-based depletion of PAF1 from nuclear extracts has allowed the in vitro RNA Pol II transcriptional recycling assay to be repeated with and without PAF1, demonstrating its requirement for proper RNA Pol II transcription from the recycled template (Fig. 3b and Supplementary Fig. 5c, d). Ser 2 phosphorylated Pol II ChIP-seq in the presence or absence of siRNA targeting PAF1 has demonstrated that PAF1-depleted cells show defective RNA Pol II recycling in the cells (Fig. 4 and Supplementary Fig. 6). Thus, this in vitro methodology is most informative when used in combination with cellular transcription studies that test both the movement of the factor of interest through the gene body and back to the transcription start site and its requirement for the movement of other factors such as RNA Pol II.

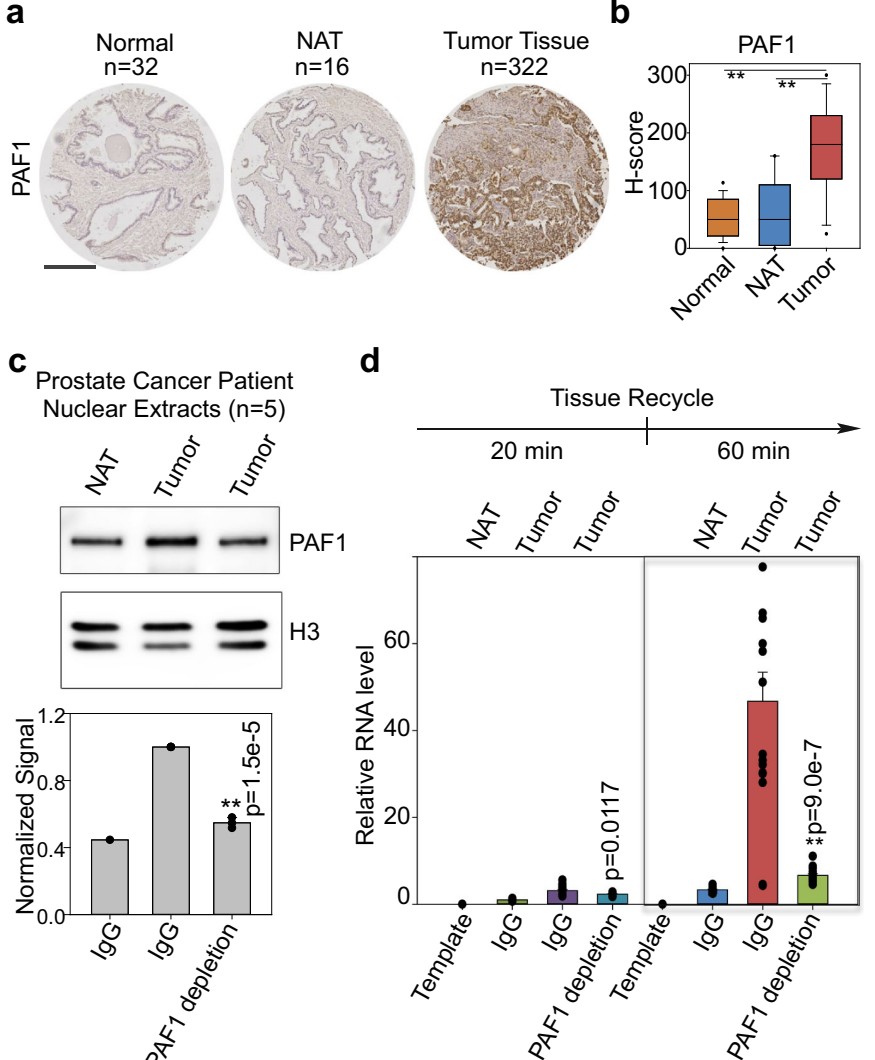

**Fig. 5 PAF1 contributes to elevated tumor recycling. a** Representative PAF1 immunoreactivity in normal prostate, non-tumor adjacent tissues (NAT), and tumor tissues. Scale bars: 500 μm. **b** Slides were scanned using an Aperio Digital Pathology Slide Scanner (Leica Biosystems) at 20× magnification. A box plot compares H-scores of PAF1 nuclear staining in 370 tissues. Box plots show median (center), 25th/75th percentiles (box bounds), 10th/90th percentiles (whiskers), and 5th/95th percentiles (outliers as symbols). The values beyond the outliers are not shown in the boxplot. The significance was determined by one-way ANOVA, **$p < 0.001$. **c** Five prostate cancer tissues with paired NAT were used. Nuclear extracts of pooled tissues were incubated with PAF1 antibodies to moderately deplete PAF1 protein and then were analyzed by western blot. Histone H3 was used as an endogenous loading control. The graph shows the quantification of PAF1 expression. Values were expressed as the mean ± SEM of three independent experiments. **d** The transcription recycling assay was performed using pooled tissue nuclear extracts with or without PAF1 depletion. In vitro transcribed RNA products from the 2nd template during the Recycle step were analyzed by reverse transcription followed by qPCR quantification. Values were expressed as the mean ± SEM of three independent experiments. *p*-values were calculated using two-tailed Student's *t*-test, **$p < 0.01$. Source data are provided as a Source data file.

The PAF1 complex is best known as an RNA Pol II interacting factor[34] that scaffolds the recruitment of additional proteins[35,36] to facilitate transcriptional elongation[14,37]. Recent studies have further found that the PAF1 complex can regulate promoter-proximal Pol II pausing, transcription termination, and chromatin structure[8]. By using an in vitro transcription system complemented by timed cellular ChIP-seq experiments, we find that PAF1 drives transcriptional recycling (Figs. 1–5). Other PAF1 complex subunits such as LEO1 may also participate in transcriptional recycling (Supplementary Fig. 4g–i, 5f, g). Although PAF1 does not appear to promote noncanonical TSS proximal polyadenylation site usage in our cell models (Supplementary Fig. 8a), analysis of our Ser2 phosphorylated Pol II timed ChIP-seq data found that PAF1 can also function to accelerate transcription elongation rates (Supplementary Fig. 8b, c). Together, both the findings from this study and

previously published data are consistent with the notion that PAF1 can regulate many aspects of Pol II transcription and that our knowledge of the molecular functions of PAF1 is in the process of rapidly expanding[8].

Application of the in vitro methodology to nuclear extracts from tumor and adjacent non-tumor patient tissues led to the detection of an abnormally high rate of RNA Pol II recycling in human prostate cancer tissues that is at least partially PAF1-dependent (Fig. 5d) and consistent with their elevated PAF1 levels (Fig. 5b). This raises the interesting possibility of a larger trend in which some tumor tissues exhibit higher levels of transcriptional recycling relative to normal tissue so that targeting recycling-associated factors such as PAF1 might be a good therapeutic strategy for cancer types that currently have no effective targeted therapy. Indeed, previous studies have reported that PAF1 overexpression promotes the development of

multiple cancer types[38–40] and correlates with adverse outcomes in non-small cell lung cancer in particular.

Additional applications of this in vitro methodology include evaluating other cell-based or tissue-based models of disease for differences in RNA Pol II transcriptional recycling relative to normal controls, using nuclear extracts, biotinylated templates, and a simple qPCR readout from Template 2. The method is also useful as a screening tool, to test the requirement for candidate proteins in transcriptional recycling. Aside from the PAF1 complex, other identified candidate recycling factors from this study remain to be individually characterized by cell-based transcription assays. Importantly, this methodology has proven to be a means of identifying functionally important factors for targeting RNA Pol II transcription at multiple levels of regulation, including at a recycling level that appears to be dysregulated in some disease states.

## Methods

**Mammalian cell lines**. The human male LNCaP-abl cell line has been used in previous studies as a model for castration-resistant prostate cancer[41,42], and was authenticated as described previously[43]. LNCaP-abl and the human benign prostatic hyperplasia cell line BPH1 (Sigma) were cultured in phenol red-free RPMI 1640 (Invitrogen) supplemented with 10% charcoal-stripped FBS (Omega Scientific) and 1mM L-glutamine. The human female HeLa cell line (American Type Culture Collection) was grown in DMEM (Invitrogen) with 10% fetal bovine serum (Omega Scientific). All cultures were incubated at 37 °C in 5% CO₂. All cell lines have been authenticated prior to commencing this study by short tandem repeat (STR) profiling and karyotyping. All cell lines were routinely tested to ensure that they were free of mycoplasma contamination (Venor™GeM Mycoplasma Detection Kit, Sigma-Aldrich).

**Human tissues**. Radical prostatectomy specimens were obtained from Duke University. Immediately after surgery, a genitourinary pathologist reviewed slides stained with hematoxylin and eosin from each case, and circled areas estimated to be >70% for prostate tumor tissue or 100% for non-malignant adjacent prostate epithelium, respectively, corresponding to the frozen sections. Using a 4 mm punch tool, tumor or non-malignant adjacent tissues were extracted from the mirrored face of the frozen section according to the pathologist's annotation, and the tissues were rapidly snap-frozen. All experimental procedures were approved by the Duke University Institutional Review Boards.

### In vitro transcription recycling assay

*Nuclear protein extraction*. The nuclear extract used in this system was either purchased from Promega or prepared as described with modifications[44]. Briefly, HeLa or LNCaP-abl cells were washed, trypsinized, and resuspended in cold PBS. After centrifugation, cell pellets were suspended in ten volumes of Buffer A (10 mM HEPES pH 7.9, 3 mM MgCl₂, 10 mM KCl and 0.5 mM DTT) and allowed to stand for 10 min. All cells were lysed by ~10 strokes of Dounce homogenizer (B type pestle) and checked by Trypan blue staining. Nuclei were collected after centrifugation for 10 min at 4,000 rpm at 4 °C and resuspended in two volumes of Buffer C (20 mM HEPES pH 7.9, 25% (v/v) glycerol, 0.42 M NaCl, 1.5 mM MgCl2, 0.2 mM EDTA, 0.5 mM DTT and 1x Protease Inhibitor Cocktail) for 40 min at 4 °C. The suspension was centrifuged for 20 min at 25,000 × g and dialyzed against 100 volumes of Buffer D (20 mM HEPES pH 7.9, 20% (v/v) glycerol, 0.1 M KCl, 0.2 mM EDTA, 0.5 mM DTT, 1x Protease Inhibitor Cocktail) for 6–12 h. The supernatant was frozen in aliquots on dry ice and stored at −80 °C. The protein concentration was usually 2−5 mg per ml. For human tissues, samples were cut into small pieces and stroked 50 times by homogenizer. After incubating on ice for 15 min and filtering by strainer, tissues were centrifuged for 10 min at 4 °C. The tissue pellet was then resuspended with Buffer C for 40 min at 4 °C with vortexing every 5 min. The protein concentration of human tissue nuclear extract was usually 1−3 mg per ml. For the immunodepletion assay, nuclear extract was incubated with IgG control or indicated antibodies for 2 h at RT or overnight at 4 °C, followed by the addition of protein A/G beads and incubating for 2 h at RT. After centrifuging, the supernatant was saved and defined as control or depleted nuclear extracts. Western blot showed that a reduction of approximately 30–50% was achieved for nuclear extract immunodepletion.

*Linear DNA template amplification and immobilization*. Linearized DNA templates were prepared by the amplification of p300 and pEF1 plasmids using 5'-biotinylated forward primers (see primer sequences in Source Data of Supplementary Fig. 1) with Q5 High-Fidelity 2X Master Mix following the manufacturer's instructions. The CMV template (from p300) includes the CMV enhancer (235–614), the CMV promoter (615–818), and the ORF (1020–8279) (Supplementary Fig. 1A). The EF1α Template (from pEF1) consists of the EF-1α promoter (2013–3191) and the ORF (3207–5615) (Supplementary Fig. 1C). Initial denaturation was performed at 98 °C for 30 s, followed by 30 cycles of 98 °C for 10 s, 56 °C for 30 s, and 72 °C for 10 min, with final extension performed at 72 °C for 10 min. Amplified templates were purified using AMPure XP beads and quantified with a DeNovix DS-11 FX + spectrophotometer (Wilmington,

DE). The Dynabeads kilobaseBINDER™ Kit was used for immobilizing long double-stranded DNA templates. 60 pmoles of biotinylated DNA fragments were immobilized onto 1 mg Dynabeads M-280 Streptavidin or Pierce™ NeutrAvidin™ Agarose (Thermo) for 3 h at room temperature. After washing in 10 mM Tris-HCl pH 7.5, 1 mM EDTA, and 2.0 M NaCl twice and Tris-HCl pH 8.0 three times, immobilized templates were resuspended in distilled RNase-free water.

*In vitro transcription and recycling*. For transcription on the 1st template, PIC formation was set up by incubating 3–5 mg of nuclear proteins with 100 pmoles of immobilized template in a final volume of 1.2 ml (10 mM HEPES pH 7.9, 10% (v/v) glycerol, 0.25 mM DTT, 0.1 mM EDTA, 50 mM KCl, 3 mM MgCl₂) for 30 min at 30 °C at 1000 rpm rotation to mix. The transcription was started by the addition of NTP mix to a final concentration of 600 μM each, and allowed to proceed for 1 h. The reaction was then stopped by placing the tube on a magnet for 30 s followed by discarding of the supernatant. Templates were washed quickly twice for 2 min in ice-cold washing buffer (10 mM HEPES pH 7.9, 50 mM KCl, 0.1 mM EDTA, 0.25 mM DTT 10% glycerol, 0.05% NP-40) and once without NP-40. Transcription on the 1st template was allowed to restart by incubating templates with new transcription buffer (10 mM HEPES pH 7.9, 10% (v/v) glycerol, 0.25 mM DTT, 0.1 mM EDTA, 50 mM KCl, 3 mM MgCl₂, 600 μM NTP mix). After 30 min, the 1st round of solutions from the 1st template was collected by separating the 1st template-coated beads with a magnet for 30 s. New transcription buffer was added back to the 1st template-coated beads and transcription was allowed to proceed for another 20 min before collection of the 2nd round of transcription solutions from the 1st template. The 1st or 2nd round of solutions from the 1st template was incubated immediately with the 2nd template to allow transcription recycling start. After incubating for 20 min or 1 h at 30 °C, solutions containing the 2nd template-derived RNA products were collected for RNA quantification. Proteins bound to the 2nd template were eluted in 1× SDS loading buffer and analyzed by gel electrophoresis. Of note, "1st template" and "2nd template" are defined differently in the gene expression analysis as opposed to the proteomics analysis. For analysis of RNA transcripts from recycling, we used the CMV template as "1st template" and the EF1α Template as "2nd template", as the RNA products from the 2nd template (i.e., RNA from recycling) must be distinguishable from those from the 1st template. For proteomics analysis of proteins associated with the templates, we used the same CMV template as "1st template" in PIC and multi-round transcription steps and "2nd template" in the recycling step (Fig. 2). Using the sample template (with the same promoter) avoids promoter bias when we compare associated proteins from different transcription steps. To alleviate concerns that these proteomics results from the CMV template might be limited to a single type of template, the experiment was reproduced using an alternative template (i.e., using EF1α template as "1st template" in PIC and multi-round transcription steps and the same template as "2nd template" in the recycling step, Supplementary Fig. 4).

*RNA quantification by qPCR*. RNA quantification of in vitro transcription was performed as described with some modifications[45]. RNA products in transcription recycling were purified using the RNeasy mini kit with On-Column DNase digestion (Qiagen), and further concentrated using the RNeasy MinElute Cleanup kit. Next, reverse transcription was performed with SuperScript IV VILO Master Mix with additional ezDNase enzyme treatment (Thermo). The DNA digestion reaction was incubated at 37 °C for 3 min followed by annealing at 25 °C for 10 min, reverse transcription of RNA at 50 °C for 10 min, and inactivation of enzyme at 85 °C for 5 min. The diluted cDNA was then used for qPCR quantification (Thermo). Primer pairs were designed for qPCR detection to cover the transcribed regions of the CMV template and the EF1α template (see primer sequences in Source Data of Supplementary Fig. 1). Standard curves were generated for each primer pair and used to calculate RNA copy number. By normalizing to the copy number of one reaction (vehicle or siControl or wild type) on the 1st template at 1 h or the 2nd template at 20 min, relative RNA level was finally determined for each experiment. For transcription from full-length templates and promoter-deleted templates, 100 ng of the CMV or EF1α full-length template or promoter-deleted template was incubated with 400 μg of HeLa nuclear extract for 30 min to form the PIC, and NTPs were added to the reaction to allow elongation to proceed. After 1 h, all reactions were stopped by adding 25 mM EDTA. RNA products were purified and analyzed by RT-qPCR.

**Two-dimensional gel electrophoresis (2D-DIGE)**. 2D DIGE was performed by Applied Biomics, Inc (Hayward, CA). Briefly, proteins bound to the DNA template were eluted in 2D lysis buffer (7 M urea, 2 M thiourea, 4% CHAPS, 30 mM Tris-HCl, pH 8.8), and protein concentration was measured by the Bio-Rad protein assay method. For protein labeling, 30 μg of each protein sample was labeled with 0.7 ml of Cy2, Cy3, or Cy5 at 4 °C for 30 min. Labeling was then stopped by the addition of 0.7 ml of 10 mM L-lysine and incubation for 15 min at 4 °C. The labeled samples were mixed together and diluted with an equal volume of 2×2D sample buffer (8 M urea, 4% CHAPS, 20 mg/mL DTT, 2% pharmalytes, and trace amount of bromophenol blue) and 100 ml DeStreak solution. After adjusting the final volume with rehydration buffer (7 M urea, 2 M thiourea, 4% CHAPS, 20 mg/mL DTT, 1% pharmalytes, and a trace amount of bromophenol blue), samples were incubated at room temperature for 10 min and centrifuged for 10 min at 16,000 × g, and then subjected to isoelectric focusing (pH 3–10) according to the GE Healthcare protocol. The IPG strips were incubated in fresh equilibration buffer #1 (50 mM Tris-HCl, pH 8.8, containing 6 M urea, 30% glycerol, 2% SDS, a trace amount of bromophenol blue, and

10 mg/ml DTT) for 15 min and subsequently rinsed in fresh equilibration buffer #2 (50 mM Tris-HCl, pH 8.8, containing 6 M urea, 30% glycerol, 2% SDS, a trace amount of bromophenol blue, and 45 mg/ml Iodoacetamide) for 10 min with gentle agitation. IPG strips were rinsed once in the SDS-gel running buffer and transferred to 8% SDS-gels, and run at 150 V at 15 °C until the dye front reached the bottom of the gel. After SDS-PAGE, gels were scanned using the Typhoon TRIO (GE Healthcare, Waukesha, WI) following the manufacturer's instructions. The images were analyzed and processed by Image Quant software (version 6.0, GE Healthcare, Waukesha, WI), and quantitation analysis was done with DeCyder software (version 6.5, GE Healthcare, Waukesha, WI).

**Nano-liquid chromatography coupled with tandem mass spectrometry (Nano-LC-MS/MS).** In this experiment, all protein samples were subjected to digestion with trypsin, and were analyzed on a Dionex Ultimate 3000 Nano LC system coupled with an Obitrap Q Exactive HF mass spectrometer (Thermo Fisher Scientific, USA) with an ESI nanospray source by Creative Proteomics (Shirley, NY) and The Sanford Burnham Prebys Proteomics Shared Resource (La Jolla, CA). Briefly, the samples were reduced by 10 mM DTT at 56 °C for 1 h and alkylated by 20 mM IAA at room temperature in the dark for 1 h. Free trypsin was added into the protein solution at a ratio of 1:50, and the solution was incubated at 37 °C overnight. Samples were cleaned up with C18 tips, and the extracted peptides were lyophilized to near dryness. Peptides were resuspended in 2–20 µl of 0.1% formic acid before LC-MS/MS analysis. 5 µl of the sample was loaded onto a 100 µm × 10 cm in-house made column packed with a reversed-phase ReproSil-Pur C18-AQ resin (3 µm, 120 Å, Dr. Maisch GmbH, Germany). Peptides were separated at an analytical flowrate of 600 nl/min with a linear gradient (A: 0.1% formic acid in water; B: 0.1% formic acid in acetonitrile): from 6 to 9% B for 8 min, from 9 to 14% B for 16 min, from 14 to 30% B for 36 min, from 30 to 40% B for 15 min and from 40 to 95% B for 3 min, eluting with 95% B for 7 min. Data-dependent acquisition was performed using the in positive ion mode. Survey spectra were acquired in the Orbitrap with a resolution of 60,000 and a mass range from 300 to 1800 m/z. Up to 20 of the most intense ions from the preview scan in the Orbitrap were isolated, fragmented, and analyzed in the LTQ part of the instrument. MS files were analyzed and searched against a human protein database based on Uni-ProtKB/Swiss-Prot (2018) using Peaks Studio X (Bioinformatics Solutions, Canada). The parameters were set as follows: the protein modifications were carbamidomethylation (C) (fixed), oxidation (M) (variable), deamidation (NQ) (variable); the enzyme specificity was set to trypsin; the maximum missed cleavages were set to 2; the precursor ion mass tolerance was set to 10 ppm, and MS/MS tolerance was 0.5 Da. Only high confidence identified peptides (FDR 1%) were chosen for downstream protein identification analysis. Label-free quantitative proteomics analysis was performed using Peaks Studio X Q module with modified parameters (fold change >2, FDR (adjusted P value) <1%, unique peptides >3).

**Chromatin immunoprecipitation (ChIP) and ChIP-seq.** ChIP-seq was performed as previously described[46]. Human LNCaP-abl cells were either not treated, or treated with flavopiridol (FP) (300 nM) or vehicle (DMSO) over a time course. 1–2 × 10^7 cells were fixed with 1% formaldehyde for 10 min at room temperature, and then washed with ice-cold PBS three times. Cells were then collected and resuspended in Lysis Buffer-Protease Inhibitor solution (1% SDS, 5 mM EDTA, 50 mM Tris-HCl pH 8.1, 1x Protease Inhibitor), on ice for 10 min. Sonication was performed using Cell Disruptors (Branson Ultrasonics™ Sonifier™) with 5–8 cycles (10 s on/30 s off on ice). After centrifuging for 10 min at 14,000 rpm at 4 °C, the supernatant was collected and immunocleared by incubating with BSA-blocked protein A/G sepharose beads at 4 °C for 1 h. Then 4–6 µg of the indicated antibodies were added into chromatin solution and incubated overnight at 4 °C on a rotator. Beads were washed successively with 1 ml of TSE I buffer (0.1% SDS, 1% Triton X-100, 2 mM EDTA, 20 mM Tris-HCl pH 8.1, 150 mM NaCl), 1 ml of TSE II buffer (0.1 % SDS, 1% Triton X-100, 2 mM EDTA, 20 mM Tris-HCl pH 8.1, 500 mM NaCl), 1 ml of Buffer III (0.25 M LiCl, 1% NP-40, 1% deoxycholate, 1 mM EDTA, 10 mM Tris-HCl pH 8.1) and TE (10 mM Tris-HCl, pH8.0, 1 mM EDTA) twice. Then, DNA–protein complexes were eluted in 100 µl of extraction buffer (10 mM NaHCO₃, and 1% SDS), and incubated overnight at 65 °C to reverse the crosslinks of protein–DNA interactions. Finally, DNA was purified with the PCR Purification Kit (QIAGEN). For high throughput sequencing, DNA libraries were generated using NEBNext® ChIP-Seq Library Prep Master Mix Set for Illumina (NEB) and sequenced using an Illumina HiSeq 2500 instrument at the Ohio State University Comprehensive Cancer Center sequencing core, and using Illumina HiSeq4000, NextSeq500, and NovaSeq6000 instruments at Duke University Sequencing and Genomic Technology Shared Resource. The experiment was performed at least three times.

**Sequencing data analysis.** Raw reads were aligned to the reference genome (hg19) using Bowtie 2 with default parameter settings. Clonal reads and bad-quality reads were removed. The remaining reads were extended by 300 bp (presumably coinciding with library size) in the 5′-3′ direction along the chromosome axis. The tracks of coverage density were made with extended reads, which were normalized to the same sequencing depth (50 million). HOMER (v4.0)[47] was used to quantify promoter read densities and calculate the pausing ratio of Pol II and PAF1 with default parameters. Briefly, to calculate the pause and release ratio, we first normalized all time-resolved ChIP-seq

datasets of Pol II (Ser2) and PAF1 to the same sequencing depth of 50 million. Each transcript was then divided into three regions, including a 5′-end bin [TSS −50 bp, TSS +200 bp], a gene body bin [TSS +200 bp, TTS −500 bp] and a 3′-end bin [TTS −500 bp, TTS +500 bp]. The average density per nucleotide was calculated using the normalized density files for each bin. The 5′ pausing ratio compares the 5′-end density to the gene body density, while the 3′ release ratio compares the 3′-end density to the gene body density. We calculated the ratios for Pol II (Ser2) and PAF1 occupancy at the indicated time points.

**RNA interference.** For transient knockdown of PAF1, ON-TARGETplus *PAF1* siRNA and ON-TARGETplus Non-targeting Pool were purchased from Dharmacon. A total of 5 × 10^6 LNCaP-abl cells were plated in 15 cm dishes and then transfected with Lipofectamine® RNAiMAX Reagent (Invitrogen) according to the protocol provided by the manufacturer. After 3 days, nuclear proteins were extracted using the NE-PER Nuclear and Cytoplasmic Extraction Kit (Thermo Scientific) according to the manufacturer's instructions.

**Western blotting.** Protein samples were boiled for 5 min and then resolved in 4–15% or 7.5% mini-PROTEAN TGX Gels (Bio-rad). Proteins were transferred to PVDF membranes using a Semi-Dry transfer cell or Trans-Blot cell (Bio-rad). Membranes were blocked with Azure Chemi Blot Blocking Buffer (Azure) at room temperature for 1 h, and then incubated with the appropriate primary antibodies at 4 °C overnight, followed by washing and incubation of secondary antibodies at room temperature for 1 h. Immunoblot signal was developed with Chemiluminescent substrate for quantitative chemiluminescent Westerns (Azure) and captured using the C-DiGit Chemiluminescent Western Blot Scanner (Li-Cor) or Azure Western Blot Imaging System (Azure). Western blots were performed at least twice.

**Immunohistochemistry (IHC) analysis.** Tumor microarrays containing normal prostate (32 samples), non-cancer adjacent tissue (NAT, 16 samples), or prostate cancer (322 samples) were purchased from US Biomax (Derwood, MD). Immunohistochemical staining of these samples was performed at Duke Department of Pathology with a PAF1 rabbit polyclonal antibody (ab137519, Abcam). Briefly, following deparaffinization, antigen retrieval was performed for 40 min using Reveal Decloaker solution (Biocare Medical), followed by 20 min cooling. This was followed by the application of Protein Block (Biocare Medical) for 15 min and Endogenous Peroxidase Quench (Biocare Medical) for 6 min. Primary antibody was applied for 60 min at a dilution of 1:200, while secondary antibody detection was performed as part of the MACH 3 detection system (Biocare Medical). Counterstaining was performed with hematoxylin. Slides were digitally scanned at 20× magnification using an Aperio digital pathology slide scanner (Leica Biosystems) by Duke Image Cytometry lab. H-scores were assigned by Drs. Yue Zhao and Jiaoti Huang of Duke University Department of Pathology. These scores ranged from 0 to 300 and were calculated as the product of the Intensity Score for the epithelial region of maximum PAF1 staining intensity in each sample (assigned on a scale from 0 to 3) multiplied by the percentage of epithelial cells in that sample showing maximum staining intensity (0–100%).

**Statistics.** All quantification and statistical analyses were performed using Sigmaplot (13.0), Partek Genomics Suite (7.18), and Peaks Studio (X). Data are shown as mean ± SEM of biological triplicates. Details of the statistical analysis for each experiment can be found in the relevant figure legends. All statistical analyses were calculated using two-tailed Student's *t*-test unless otherwise mentioned.

**Reporting summary.** Further information on research design is available in the Nature Research Reporting Summary linked to this article.

## Data availability

The data that support this study are available from the corresponding authors upon reasonable request. The high throughput sequencing data generated in this study have been deposited in the Gene Expression Omnibus database under accession number GSE133655. The Proteomics data generated in this study have been deposited in the PRIDE database under accession numbers PXD027143 and PXD017881. Source data are provided with this paper.

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

## Acknowledgements

This work was supported by NIH grants R01 CA200853 (Q.W.), U54 CA217297 (Q.W.), R01 GM120221 (Q.W.), R01 CA234575 (R.G.R.) and R50 CA251843 (Z.C.).

## Author contributions

Z.C. and Q.W. designed research. Z.C. performed most of the experiments with assistance from W.H., Y.Z., J.G., F.H., H.W., and A.R.C. Z.C. performed bioinformatics analyses. Z.C., R.G.R., J.H., and Q.W. analyzed data; Z.C., W.H.,, and Q.W. wrote the paper with input from J.H. and R.G.R.

## Competing interests

J.H. is a consultant for or owns shares in the following companies: Kingmed, MoreHealth, OptraScan, Genetron, Omnitura, Vetonco, York Biotechnology, Genecode, VIVA Biotech, and Sisu Pharma, and received grants from Zenith Epigenetics, BioXcel Therapeutics, Inc., and Fortis Therapeutics. The other authors declare no competing interests.
