## [Peer Review File · Nature Communications]

Transcription recycling assays identify PAF1 as a driver for RNA Pol II recyclingReviewers' Comments:

Reviewer #1:

Remarks to the Author:

Wang et. al have identified PAF1 complex components as important factors contributing to RNA Pol II recycle using an immobilized two-templates system, and have revealed the clinical relevance to prostate cancer. This in vitro system enables the isolation and identification of proteins involved in each transcriptional step. The authors have taken an interesting approach addressing an important long-standing question in the field. Here are a few comments:

1. One wonders if the "recycled factors" from templates 1 is sufficient to start transcription of templates 2, the authors also discussed this methodology, which appears not to convincingly distinguish "drop-off" and "run-off" factors.
2. For the last sentence on Page 8, "PAF1 occupancy over 5000 human RefSeq genes in LNCaP-abl cells (Supplementary Fig. 3E) showed enrichment at transcription start sites (TSS) and absence from the gene body following 1 hr FP treatment, indicating that PAF1 returns to the TSS following transcription termination in vivo." Since PAF1 complex can regulate Pol II pausing, elongation and termination, there's no evidence to suggest "PAF1 returns to the TSS following transcription termination in vivo".
3. Ali Shilartifard's group reported that PAF1 inhibit paused Pol II release at promoter proximal (reference 18), thus the reduced Pol II occupancy at TSS could be explained by increased Pol II release but not by "decreased recycle" when PAF1 was knocked down (Fig 5).
4. For the conclusion that PAF1 functions with other subunits of the PAF1 complex (PAF1C) to promote the transcription recycling, it means that the intact complex which includes PAF1, LEO1 and WDR61 that functions in the POI II recycling. To verify this claim, at least a couple of other subunits of the PAF1 complex should be deleted respectively while PAF1 is intact to see the effects on transcription recycling.
5. "Production of RNA by the proteins contained in the first elution did not change significantly with PAF1 depletion, indicating that transcriptional machinery was sufficiently recycled to permit some continued transcription. The second elution, however, exhibited a significant decrease in transcriptional activity from the IgG control to each antibody-depleted condition, indicating that PAF1 removal significantly attenuates transcriptional recycling." One wonders (1) why transcription of the first elution is not affected by PAF1-deletion at all, and (2) if a third elution was added, then what might happen to the transcription recycling.
6. The authors should discuss if the "recycling" role of PAF1 may or may not agree with the observation that the knock-down of the PAF1 speeds up the transcription elongation rate.

Reviewer #2:

Remarks to the Author:

Chen et al. propose a two template -based assay for investigating proteins that aid in RNA Polymerase II recycling, which is an important mechanism for regulating gene expression. Chen et al. claim the novel aspect of their assay is its ability to distinguish between "cycling" and "recycling" and that the assay does not specifically capture proteins that aid in re-initiation, which the authors seem to treat as separate from recycling. However, in this reviewer's opinion the authors fail to establish a clear and biologically relevant difference between "cycling" and "recycling" in their in vitro assay. Furthermore, the authors claim to be able to accurately collect transcription complexes at specific phases of transcription at different time points, but do not provide any details on how they experimentally

validated these collections. These flaws, combined with low replicate number (n=2), when included, leave the reviewer unable to conclude any biological or statistical significance of this assay. The lack of validation and experimental clarity continued in the cell based experiments. Replicate number was not included for any of the ChIP-seq experiments, and detailed information on how the authors calculated their "Pausing Ratio" was not included in the manuscript. Additionally, these experiments were done in a prostate cancer cell line, when the authors later go on to show elevated Paf1 levels in prostate tumor samples and argue it's biologically relevant, but there was no discussion of relative Paf1 levels in this cell line and how that could affect these ChIP experiments. Furthermore, there was no experimental data validation the siRNA used to knockdown Paf1 in flavopiridol (FP) ChIP experiments. Finally, an antibody against RNAPII phosphorylated on Ser2 of the C-terminal domain was used in the FP ChIP experiments, but the mechanism of FP is inhibiting this exact phosphorylation mark, causing an inherent bias against pulling down Ser2 phosphorylated RNAPII after FP treatment. In conclusion, while the data likely shows that Paf1 is required for proper RNAPII elongation and can thereby contribute to proper gene expression, but the reviewer is left unable to agree with claims of Paf1 driving RNAPII recycling. The overall lack of rigor and replicates (or at least replicate definition) are significant concerns.

Specific Comments:

1. For the in vitro assay, the authors very specifically differentiate "cycling" as transcription rounds on the same template, and "recycling" as transcription rounds on a new, second template. However, this differentiation seems to disappear in vivo as "recycling" is then talked about in the context of multiple rounds of transcription on one gene. The authors fail to establish how these two processes they distinguish between for data collection and analysis in their in vitro assay, and which seem to be they key novel factor in their assay, translate in vivo. Relatedly, in the introduction, the authors seem to establish re-initiation as a separate process from what they define as recycling, which does not seem clear to this reviewer, as re-initiation is key for successful sequential rounds of transcription.
2. In the in vitro assay, the authors assume they are collecting RNAPII complexes at very specific phases of transcription, but no clear details or validation is provided to explain their rationale for knowing which complex is which at these specific timepoints. This uncertainty leaves the reviewer unsure of the validity of this assay.
3. Figure 1. "Regions = 7" in the legend is not defined. Low replicate number of n=2 lacks statistical power, as none of the points in 1b are denoted as significantly different.
4. Figure 2. 2b is described as "Comparison of the proteins detected in each phase of transcription" but is not accurate as not all three phases are compared together, two 1:1 comparison are made. There is no discussion in the figure legend of replicate number, or how "overrepresentation" and "underrepresentation" were actually quantified from the LC-MS/MS data. There also seems to be much more overlap between the two phases than there are differences, again leaving the reviewer to wonder how these complexes were known to be collected as specific phases. 2d plot does not have a y-axis labels (same in Figure 3C)
5. Figure 4. 4A does not have a molecular weight marker. Also, there seem to be multiple Paf1 bands, or if they are not all supposed to be Paf1, which one it should be is not clearly indicated (antibody specificity should be discussed). If all are thought to be Paf1 some discussion on if they're cleavage or degradation products would be helpful, as well as why the authors think the protein is being degraded to such an extent in their samples. While authors claim moderate Paf1 depletion, without quantitation it seems like there is still quite a lot of Paf1 left. Quantitation is required with at least three biological replicates to make this type of statement.
6. Figure 5 and included experiments. ChIP-seq was performed using an antibody against Ser2 phosphorylated RNAPII on samples treated with flavopiridol (FP), and then let to recover after FP release. However, the mechanism of FP is an inhibitor of PTEFb/Cdk9 kinase which is responsible for phosphorylating RNAPII on Ser2. Therefore FP treatment creates a confounding factor in these ChIP experiments that the reviewer feels nullifies any major conclusions that could be made. Additionally, the authors provide no validation data for the siRNA to show knockdown used in these same experiments, and no replicate number is included. The wide standard deviations and lack of an indication of significance in 5e lead the reviewer to believe low replicate number. Finally, any details

on how the Pausing Ratio is calculated for these data are not included in this manuscript.

7. Figure 6. In 6C the authors again provide no quantitation of Paf1 depletions, and visually/qualitatively it looks modest at best. 6a/b does seem to show elevated Paf1 levels in prostate tumor samples. However, this conclusion creates a confounding factor for any experiments where the prostate cancer cell line (LNCaP-abl) was used, as it seems likely this cell line could also then have high Paf1 levels. However, this possible limitation was not discussed or explored.

8. Altogether, the data doesn't seem strong enough or direct enough to conclude that Paf1 is a driver of recycling. It does seem likely that Paf1 is required for full efficiency of transcription elongation and that depleting it leaves RNAPII more open to alternative termination mechanisms before reaching the canonical termination site. However, this manuscript does not seem to consider that possibility, or any other data-based alternatives when presenting the model of Paf1 as a driver of recycling. Overall, the lack of rigor and reproducibility in these experiments, still leaves the reviewer unable to agree with any significant conclusions made from these data.

Minor points

1. Supplementary Figure 4 is used to conclude that Leo1 and WRD61 depletion do not exacerbate the effect of Paf1 depletion. The authors did not use proper control of individually depleting Leo1 or WRD61.

Reviewer #3:

Remarks to the Author:

This manuscript presents a novel methodological approach for assessing different stages of gene regulation. Importantly, this permits the analysis and comparison between expression from the first template, versus expression from subsequent templates. The authors show that the protein machinery from the first template was capable of regulating expression from the second template, which implies that RNA Pol2 is recycled. This method uses an in vitro template coupled with cell lysates, permitting the discovery of known and unknown components of the process. The authors couple this capture approach with 2D gels and Mass Spec to identify proteins involved in the different stages of gene expression, revealing a number of known factors and confirming that discrete complexes exist. By exploring proteins that are enriched in the recycling phase, the authors identify the PAF1 complex as being significantly biased towards the second template, suggesting that the PAF1 complex is involved in the recycling process. A series of ChIP-seq experiments are undertaken to show that PAF1 is recycled to the promoters of target genes in a prostate cancer cell line model. Data is provided to show that depletion of PAF1 doesn't impact expression from the first template, but has profound consequences on expression from the second template, confirming a functional role for the PAF1 complex in recycling. This is further validated by assessing RNA Pol2 recycling in growing cells that have been depleted for PAF1. Finally, PAF1 is shown to be overexpressed in prostate cancer relative to adjacent normal or healthy tissue.

This is a very good manuscript. The method is novel and is exploited in a smart and effective manner, revealing new insight into the fundamentals of gene regulation and complex recycling. The paper is well written and the data/figures are clear and convincing. I have a few issues that need to be addressed, but I think this would be an important contribution to the scientific community.

- It wasn't clear (even after wading through the methods section) how many replicates were conducted of the Mass Spec discovery, how replicates were integrated or what the degree of variability is. Can the authors do a sample swap (i.e. purposely match up samples from different experimental conditions as 'replicates') to identify the degree of variability in the system? Given this is a new method, some confidence in the number of proteins that are 'differential' by chance would be an important piece of information.

- It wasn't clear why the authors did the initial discovery in HeLa cells but then jumped into the LNCaP-abl cells? What was the logic behind this and based on the final IHC data, is this the right

model to be making conclusions about PAF1 function, given the inference that PAF1 levels will be overexpressed and not representative of physiologic levels (or potentially function).

- To this point, what is the relevance of overexpressed PAF1 in cancer versus normal and will this impact the efficiency of the second template activity that might only be observed in cancer models and not representative of normal gene expression properties in non-cancerous contexts?

- Is there any published ChIP-seq data for PAF1, LEO1 and WDR61? Do they always co-bind or is there a potential role for these proteins independently of the other two proteins?

- Given the evidence that PAF1 siRNA dramatically affects cell viability, how do the authors get confidence that the changes in RNA Pol2 recycling are because of changes in recruitment to promoters, as opposed to indirect consequences of decreases in general cell viability and health?

Minor points

- The authors claim that the ChIP-seq measures 'in vivo' activity. I suspect that people who study cancer biology in animal models might take exception to this.

- The authors claim that PAF1 could be a relevant drug target because 'PAF1 as a potentially favorable therapeutic target due to its categorization as a context-specific cell fitness gene that is unlikely to perform essential functions in normal cells and may therefore be associated with less side effects'. I think it's premature to make this statement. Analysis of the Broad/Novartis DepMap suggests that 715 out of 726 cancer cell lines are dependent on PAF1. So, either PAF1 is required in almost all cancer cells, but in no normal tissue, which would require a systematic analysis of PAF1 dependence in a panel of normal cell models. Or more likely, PAF1 is required in all models, regardless of tissue of origin, whether the model is normal or diseased etc, in which case, PAF1 is unlikely to be a bone fide therapeutic target.

Response to Reviewers' Comments:

Reviewer #1 (Remarks to the Author):

Wang et. al have identified PAF1 complex components as important factors contributing to RNA Pol II recycle using an immobilized two-templates system, and have revealed the clinical relevance to prostate cancer. This in vitro system enables the isolation and identification of proteins involved in each transcriptional step. The authors have taken an interesting approach addressing an important long-standing question in the field.

Response: We thank the reviewer for appreciating the novelty and impact of our study.

Here are a few comments:

1. One wonders if the “recycled factors” from templates 1 is sufficient to start transcription of templates 2, the authors also discussed this methodology, which appears not to convincingly distinguish “drop-off” and “run-off” factors.

Response: As shown in **revised Fig. 1c**, both the first and second elutions that captured factors from Template 1 efficiently drove transcription from Template 2, while the third elution failed to continuously drive transcription from Template 2. These results suggested that, at the beginning of transcription recycling process, recycled factors from Template 1 are sufficient to start transcription of Templates 2 without adding any new transcription components. However, after two rounds of elution, the recycled factors are not sufficient to start transcription of Template 2, which may be caused by missing and/or inactivation of some necessary components.

We agree with the reviewer that our *in vitro* system is not able to distinguish “drop-off” and “run-off” factors. This limitation has been discussed in the 3rd paragraph in the “Discussion” section. Nonetheless, coupling this *in vitro* system with cellular transcription assays (timed ChIP-seq) provides a powerful means to identify and characterize putative factors participated in transcription recycling, consisting of both “drop-off” and “run-off” factors. Indeed, our timed ChIP-seq assays have established PAF1 as a run-off factor that travels through the gene body but not a drop-off factor that drops from the gene promoter (Fig. 2c, Supplementary Fig. 4e in the revised manuscript). For additional detail, please see the response to point 2 below.

2. For the last sentence on Page 8, “PAF1 occupancy over 5000 human RefSeq genes in LNCaP-abl cells (Supplementary Fig. 3E) showed enrichment at transcription start sites (TSS) and absence from the gene body following 1 hr FP treatment, indicating that PAF1 returns to the TSS following transcription termination in vivo.” Since PAF1 complex can regulate Pol II pausing, elongation and termination, there’s no evidence to suggest “PAF1 returns to the TSS following transcription termination in vivo”.

Response: We appreciate the opportunity to clarify the point we intended to make with that sentence. The reviewer is correct that we do not have direct evidence to demonstrate that “PAF1 returns to the TSS following transcription termination *in vivo*”. However, previous studies have found that flavopiridol (FP) treatment compromises new PAF1 recruitment, new phosphorylation of Pol II, and Pol II escape from the paused state, yet does not affect existing Ser2 phosphorylated Pol II (associated with PAF1) traveling within the gene body (Yu et al. *Science*, 350:1383-1386, 2015; Jonkers, et al. *Elife*, **3**, e02407, 2014; Jonkers and Lis, *Nature Reviews Mol Cell Biol*, 16:167-177, 2015). Thus, following FP treatment for 1 h, the accumulated PAF1 around the TSS will very likely have originated from traveled and recycled PAF1, as other sources have been effectively blocked. Since we do not have direct evidence to demonstrate that “PAF1 returns to the TSS following transcription termination *in vivo*”. we have revised the text to more accurately state that “*PAF1 occupancy over 5000 human RefSeq genes in LNCaP-abl cells (Supplementary Fig. 4e) showed enrichment at transcription start sites (TSS) and absence from the gene body following 1 h FP treatment. Since FP treatment compromises new PAF1 recruitment, new Pol II phosphorylation and Pol II escape from the paused state, but does not affect existing Ser2 phosphorylated Pol II (associated with PAF1) traveling within the gene body, our timed PAF-1 ChIP-seq results suggest that PAF1 may travel and recycle back to the TSS.*” (see manuscript page 8).

3. *Ali Shilartifard’s group reported that PAF1 inhibit paused Pol II release at promoter proximal (reference 18), thus the reduced Pol II occupancy at TSS could be explained by increased Pol II release but not by “decreased recycle” when PAF1 was knocked down (Fig 5).*

Response: We understand the reviewer’s idea to compare our finding to Dr. Shilartifard’s report, that PAF1 knockdown increases poised Pol II release into gene body and thus decreases poised Pol II around TSS (Chen et al. *Cell*, 162: 1003-1015, 2015). However, we would like to point out a key experimental difference, in that our timed ChIP-seq in the presence or absence of PAF1 was performed with an antibody to Ser2 phosphorylated Pol II (**Pol II [Ser 2]**). While our original Fig. 5 (Fig. 4 in revised manuscript and **New Fig. 1b**) showed a reduced Pol II (Ser 2) occupancy at TSS after PAF1 knockdown, this observation could not be explained by the release of poised Pol II from the TSS, as poised Pol II is Ser 5 phosphorylated Pol II (**Pol II [Ser 5]**) (Boehm et al. *Mol Cell Biol*, 23: 7628-7637, 2003; Kwak and Lis. *Annu Rev Genet*, 47:483-508, 2013). Instead, we believe that the reduced Pol II Ser 2 signal around TSS in PAF1 silenced cells compared with control cells in our timed Pol II Ser 2 ChIP-seq assays indicates a marked reduction of elongating Pol II (i.e. Pol II Ser 2) recycling to the TSS.

4. *For the conclusion that PAF1 functions with other subunits of the PAF1 complex (PAF1C) to promote the transcription recycling, it means that the intact complex which includes PAF1, LEO1 and WDR61 that functions in the POI II recycling. To verify this claim, at least a couple of other subunits of the PAF1 complex should be deleted respectively while PAF1 is intact to see the effects on transcription recycling.*

Response: We appreciate the reviewer’s good suggestion. We have now depleted either WDR61 or LEO1 to test the role of these PAF1 complex subunits in transcription recycling. Depletion of PAF1 or LEO1 but not WDR61 caused depletion of other PAF complex subunits and led to decreased transcription recycling (**New Supplementary Fig. 5f, g**). Our results indicate that the

PAF1 complex containing structurally and genomically tightly associated PAF1 and LEO1 (Chu et al. *Nucleic Acids Res*, 41: 10619-10629, 2013 and **New Supplementary Fig. 4g-i**) is important to enable Pol II transcription to recycle, while WDR61 is indispensable for transcription recycling. We have revised our conclusion in the paragraph entitled “PAF1 depletion attenuates RNA Pol II transcriptional recycling *in vitro*” in the Results section (Page 10).

5. *“Production of RNA by the proteins contained in the first elution did not change significantly with PAF1 depletion, indicating that transcriptional machinery was sufficiently recycled to permit some continued transcription. The second elution, however, exhibited a significant decrease in transcriptional activity from the IgG control to each antibody-depleted condition, indicating that PAF1 removal significantly attenuates transcriptional recycling.” One wonders (1) why transcription of the first elution is not affected by PAF1-deletion at all, and (2) if a third elution was added, then what might happen to the transcription recycling.*

Response:

- (1) The first elution contains higher amount of proteins that transcribe more RNA products than the second elution does. We speculate that PAF1 and some highly enriched factors in the first elution play redundant roles in the regulation of recycling on naked DNA templates. The second elution, containing a reduced amount of proteins, can be used to study functions of recycling factors with greater sensitivity. Importantly, the recycling factors discovered from functional studies using the second elution, exemplified by PAF1, play important roles in driving Pol II recycling in cells (see Fig. 4 in the revised manuscript and the above Response to Point 3).
- (2) As we have shown in **revised Fig. 1c**, the third elution is unable to continuously drive Template 2 transcription, suggesting that Pol II recycling is not an unlimited transcription process.

6. *The authors should discuss if the “recycling” role of PAF1 may or may not agree with the observation that the knock-down of the PAF1 speeds up the transcription elongation rate.*

Response: To determine the role of PAF1 in regulating transcription elongation rate, we have calculated elongation rates using the method described by Hou et al. (*PNAS*, 116: 14583-14592, 2019) with minor modifications. We selected 4,600 expressed genes (RPKM>0.3) with gene lengths >70 kb. These genes were also required to have both Pol II and PAF1 signals over the gene bodies. In order to exclude recycling signals around transcription start sites (TSS), Pol II ChIP-seq reads were divided into 1 kb bins for the region from +20 kb to +60 kb downstream of the TSS. The signal from the 10 min time point was subtracted from the corresponding signals for the 30 min and 40 min time points, respectively. After normalization of time points, a two-state hidden Markov model implemented using the depmixS4 R package was used to assign a state to each bin and to identify the transition point between Pol II-occupied and Pol II-depleted states. Elongation rates were calculated using the distances Pol II traveled at the 30 and 40 min time points. Our results indicate that knock-down of PAF1 decreases (rather than speeds up) the transcription elongation rate (**New Supplementary Fig. 8b,c**). This is consistent with previous findings that loss of Paf1 results in Pol II elongation rate defects with significant rate

compression (Hou et al. *PNAS*, 116: 14583-14592, 2019). Furthermore, our findings that PAF1 drives transcription recycling and accelerates transcription elongation rates support the notion that PAF1 can regulate many aspects of Pol II transcription and that the field is in still the process of discovering new molecular functions of PAF1 (Van Oss et al., *Trends Biochem Sci* 42, 788-798, 2017). In response to the reviewer's suggestion, we have revised our discussion on the role of PAF1 in transcription regulation in the "Discussion" section (pages 15-16).

Reviewer #2 (Remarks to the Author):

Chen et al. propose a two template -based assay for investigating proteins that aid in RNA Polymerase II recycling, which is an important mechanism for regulating gene expression. Chen et al. claim the novel aspect of their assay is its ability to distinguish between "cycling" and "recycling" and that the assay does not specifically capture proteins that aid in re-initiation, which the authors seem to treat as separate from recycling. However, in this reviewer's opinion the authors fail to establish a clear and biologically relevant difference between "cycling" and "recycling" in their in vitro assay.

Response: Please see our response below to Specific Comment 1.

Furthermore, the authors claim to be able to accurately collect transcription complexes at specific phases of transcription at different time points, but do not provide any details on how they experimentally validated these collections.

Response: Please see our response below to Specific Comment 2.

These flaws, combined with low replicate number (n=2), when included, leave the reviewer unable to conclude any biological or statistical significance of this assay.

Response: Please see our responses below to Specific Comments 3-7.

The lack of validation and experimental clarity continued in the cell based experiments. Replicate number was not included for any of the ChIP-seq experiments, and detailed information on how the authors calculated their "Pausing Ratio" was not included in the manuscript.

Response: Please see our response below to Specific Comment 6.

Additionally, these experiments were done in a prostate cancer cell line, when the authors later go on to show elevated Paf1 levels in prostate tumor samples and argue it's biologically relevant, but there was no discussion of relative Paf1 levels in this cell line and how that could affect these ChIP experiments.

Response: Please see our response below to specific comment 7.

Furthermore, there was no experimental data validation the siRNA used to knockdown Paf1 in flavopiridol (FP) ChIP experiments.

Response: Please see our response below to Specific Comment 6.

Finally, an antibody against RNAPII phosphorylated on Ser2 of the C-terminal domain was used in the FP ChIP experiments, but the mechanism of FP is inhibiting this exact phosphorylation mark, causing an inherent bias against pulling down Ser2 phosphorylated RNAPII after FP treatment.

Response: Please see our response below to Specific Comment 6.

In conclusion, while the data likely shows that Paf1 is required for proper RNAPII elongation and can thereby contribute to proper gene expression, but the reviewer is left unable to agree with claims of Paf1 driving RNAPII recycling. The overall lack of rigor and replicates (or at least replicate definition) are significant concerns.

Response: Please see our response below to Specific Comment 8. Please note that all concerns about replicates have been resolved (See our responses to Specific Comments 3-7).

Specific Comments:

1. For the in vitro assay, the authors very specifically differentiate “cycling” as transcription rounds on the same template, and “recycling” as transcription rounds on a new, second template. However, this differentiation seems to disappear in vivo as “recycling” is then talked about in the context of multiple rounds of transcription on one gene. The authors fail to establish how these two processes they distinguish between for data collection and analysis in their in vitro assay, and which seem to be they key novel factor in their assay, translate in vivo. Relatedly, in the introduction, the authors seem to establish re-initiation as a separate process from what they define as recycling, which does not seem clear to this reviewer, as re-initiation is key for successful sequential rounds of transcription.

Response: We appreciate the opportunity to clarify the important terms “cycling”, “recycling” and “reinitiation” used in our manuscript.

(1) “cycling”, “recycling” and “reinitiation” in cells.

After the initial transcription cycle that includes initiation, elongation and termination, Pol II can repeatedly transcribe the same gene and generate multiple RNA copies from the DNA template, contributing to robust overall transcriptional output (Dieci and Setenac *Trends Biochem Sci* 28: 202-209, 2003; Arimbasseri et al. *Transcription*, 5:e27639, 2014). This repeated transcription process in the cells can be called “recycling” or “cycling” (i.e. these two terms have the same meaning). The “reinitiation” is just the first step of the “recycling” or “cycling” process.

(2) “cycling”, “recycling” and “reinitiation” in our two-template-based in vitro transcription system.

In our two-template *in vitro* transcription assay, we first incubated biotinylated 1st DNA template with HeLa nuclear extracts. The preinitiation complex (PIC) was formed by incubating bead-immobilized template DNA with nuclear proteins, while the subsequent addition of nucleoside triphosphate (NTP) mix was required to trigger transcription progression. Following a 60-min of repeated transcription process (we called this

“cycling” in our *in vitro* assay), 1st template was washed and its associated factors were isolated and incubated immediately with a second DNA template (2nd template) without adding any new transcription components (i.e. nuclear extract, proteins) to 2nd template. We called the repeated transcription on the 2nd template as “recycling”, which is driven by transcription proteins from 1st template including run-off proteins from the 1st template gene body and drop-off proteins from 1st template promoter. **We think that this “recycling” process *in vitro* recapitulates the “recycling”/ “cycling” process in the cells, as both “recycling” *in vitro* and in the cells reuse transcription components (Pol II and other factors such as PAF1) from a transcribed DNA template on new rounds of transcription.** Importantly, we have demonstrated that the PAF1 protein, identified from the *in vitro* “recycling” phase, drives Ser 2 phosphorylated Pol II “recycling” in the cells (Fig. 4 in the revised manuscript). With regard to “reinitiation” *in vitro*, it is just the first step of the “recycling” process *in vitro*. As such, the “recycling” proteins we captured include proteins involved in “reinitiation”.

We recognize the shortcomings in our previous description of the *in vitro* system, and we apologize for the use of the terms “cycling” (in reference to the first template) and “recycling” (in reference to the second template). **We have now used the term “multi-round transcription” to replace the term “cycling” on the 1st template.** Please note that although “multi-round transcription” on the first template also reused transcription components from a transcribed DNA for new rounds of transcription, technically we are not able to isolate those “reused” proteins (i.e. “recycled” proteins) from a mixture of “used” and “reused” proteins in “multi-round transcription” on the 1st template. This inability to distinguish between “used” and “reused” factors during subsequent multi-round transcription on the first template was the motivation for the use of a second template.

We have also revised the Introduction part to clarify that “reinitiation” is a part of “recycling” (page 3).

2. In the in vitro assay, the authors assume they are collecting RNAPII complexes at very specific phases of transcription, but no clear details or validation is provided to explain their rationale for knowing which complex is which at these specific timepoints. This uncertainty leaves the reviewer unsure of the validity of this assay.

Response: We appreciate the opportunity to explain our rationale for designing our *in vitro* assay to collect Pol II complexes at each of the four specific transcriptional phases below:

- (1) **Pol II complex at the PIC phase:** PIC formation was set up by incubating nuclear proteins (from HeLa nuclear extract) with immobilized 1st DNA template for 30 min (Yu et al. *Nature*, 408:225-229, 2000; Hawley and Roeder. *J Biol Chem*, 262: 3452-3461, 1987), and we collected the Pol II complex bound to the template. As expected, unphosphorylated Pol II (the lower band in the total Pol II Western Blot gel, Fig. 1b left panel) and Ser2 phosphorylated Pol II were highly enriched and depleted from the PIC, respectively (**New Fig. 1b** and Sainsbury et al. *Nature Reviews Mol Cell Biol*, 16:129-143, 2015). In addition, TBP, TAFs and Mediator were highly enriched in the PIC phase (**New Supplementary Fig. 3b, c** and Sainsbury et al. *Nature Reviews Mol Cell Biol*, 16:129-143, 2015).

- (2) **Pol II complex at the Initial Transcription phase:** As we changed the term “cycling” to “Multi-Round Transcription” (see reponse to Specific Comment 1 above), we have changed to term “Elongation” to “Initial Transcription”. The Initial Transcription Phase was started by addition of NTP mix and allowed to proceed for 3 min (Yu et al. *Nature*, 408:225-229, 2000), and then we collected the Pol II complex bound to the template. Since the elongation rate of Pol II is about 2 kb/min (Danko et al. *Mol Cell*, 50:212-222, 2013; Jonkers et al. *Elife*, 3: e02407, 2014) and our DNA template is about 10 kb, the addition of NTP for 3 min only allows the initial transcription to occur. As expected, the Ser2 phosphorylated Pol II became enriched on the template during the Initial Transcription phase compared with the PIC phase (**New Fig. 1b** and Jonkers and Lis. *Nature Reviews Mol Cell Biol*, 16:167-177, 2015).
- (3) **Pol II complex at the Multi-Round Transcription phase:** For the Multi-Round Transcription phase, transcription was started by addition of NTP mix and allowed to proceed for 1 h (Yu et al. *Nature*, 408:225-229, 2000). Based on the elongation rate of Pol II and the length of our template discussed above, Pol II is able to transcribe the template multiple times during this incubation. A massive increase of Ser 2 phosphorylated Pol II but not hypophosphorylated Pol II was observed at the Multi-Round Transcription phase (**New Fig. 1b**). An increase of elongation and termination factors (e.g. SUPT6H, TOP2B and XRN2; Jonkers and Lis. *Nature Reviews Mol Cell Biol*, 16:167-177, 2015; Porrua and Libri. *Nature Reviews Mol Cell Biol*, 16:190-202, 2015) and a decrease of PIC factors (e.g. TBP, TAF2 and MED6) were also observed during this phase (**New Supplementary Fig. 3b, c**).
- (4) **Pol II complex at the Recycling phase:** **The main goal of our *in vitro* system is to characterize the Pol II complex at the Recycling phase.** As described in our earlier Response to Specific Point 1, following the multi-round transcription phase, 1st template was washed and its associated factors were isolated and incubated immediately with a second DNA template (2nd template) without adding any new transcription components (i.e. nuclear extract, proteins) to the 2nd template. Our *in vitro* “Recycling” phase served to recapitulate the “recycling”/ “cycling” process from cells, because “recycling” both *in vitro* and in cells reused Pol II transcription complex from a transcribed DNA template for new rounds of transcription. By integrating *in vitro* transcription assays with timed ChIP-seq in the cells, we found that PAF1 is required to drive Ser2 Pol II recycling (**New Fig. 1b, New Fig. 2b, revised Fig. 3, Fig. 4, revised Fig. 5d, New Supplementary Fig. 5c, d, revised Supplementary Fig. 5e, New Supplementary Fig. 5f, g, Supplementary Fig. 6**).

In summary, we believe that the validity of our *in vitro* assay is strongly supported by evidence from our own experimental data and corresponding reports in the literature.

3. Figure 1. “Regions = 7” in the legend is not defined. Low replicate number of $n=2$ lacks statistical power, as none of the points in 1b are denoted as significantly different.

Response: “Regions = 7” in the original Fig. 1 legend was defined as 7 transcribed regions within the EF1 α template (i.e. the 2nd template; see PCR primers for these 7 regions in Source Data for Supplementary Figure 1). To ease the concern of about the low replicate numbers, we have performed three independent experiments and conducted statistical analyses (please see

revised Fig. 1c, where values were expressed as the mean \pm SEM of three independent experiments).

4. Figure 2. 2b is described as ‘Comparison of the proteins detected in each phase of transcription’ but is not accurate as not all three phases are compared together, two 1:1 comparison are made. There is no discussion in the figure legend of replicate number, or how ‘overrepresentation’ and ‘underrepresentation’ were actually quantified from the LC-MS/MS data. There also seems to be much more overlap between the two phases than there are differences, again leaving the reviewer to wonder how these complexes were known to be collected as specific phases. 2d plot does not have a y-axis labels (same in Figure 3C)

Response: To address the reviewer’s concerns regarding the number of replicates for our previous proteomics assays and the lack of subsequent analysis of samples based on those replicates, we have conducted new biological triplicates with technical triplicates for each condition. We have generated a total of 36 new samples for LC-MS/MS (i.e. 9 samples for each transcription phase as described above in Response to Specific Comment 2). As shown in **New Supplementary Fig. 2a-c**, the sample correlation coefficient among both biological triplicates and technical triplicates is 0.95~0.97. Label-free quantitative proteomics analysis was performed using Peaks Studio X Q module with modified parameters (Fold change >2, FDR (adjusted P value) <1%, unique peptides >3). **Revisions to the Methods section can be found on manuscript page 24**. Unsupervised clustering analysis of these samples was applied to identify differentially enriched proteins for each specific phase (**New Fig. 2b, New Supplementary Fig. 2a-f, New Supplementary Fig. 3b**). Additionally, we have performed sample swapping analyses as suggested by Reviewer 3 (**New Supplementary Fig. 2e, f**) and demonstrated that the differential proteins we identified between phases are convincingly reproducible and were not identified by chance.

5. Figure 4. 4A does not have a molecular weight marker. Also, there seem to be multiple Paf1 bands, or if they are not all supposed to be Paf1, which one it should be is not clearly indicated (antibody specificity should be discussed). If all are thought to be Paf1 some discussion on if they’re cleavage or degradation products would be helpful, as well as why the authors think the protein is being degraded to such an extent in their samples. While authors claim moderate Paf1 depletion, without quantitation it seems like there is still quite a lot of Paf1 left. Quantitation is required with at least three biological replicates to make this type of statement.

Response: The multiple bands in the original Figure 4A were caused by western blot detection of the antibodies used for IP depletion of PAF1. Our new validation of the PAF1 antibody with siRNA knockdown found only one major specific PAF1 band at ~80 Kda (**New Supplementary Fig. 6a**). Thus, our revised figure (Fig. 3a in the revised manuscript) only shows the specific PAF1 band at 80 Kda (see full gels in Fig. 3a “Source Data”). As suggested by the reviewer, we have provided quantitation from triplicate PAF1 depletion experiments (**New Fig. 3a bottom panel and New Supplementary Fig. 5a**).

6. Figure 5 and included experiments. ChIP-seq was performed using an antibody against Ser2 phosphorylated RNAPII on samples treated with flavopiridol (FP), and then let to recover after FP release. However, the mechanism of FP is an inhibitor of PTEFb/Cdk9 kinase which is

responsible for phosphorylating RNAPII on Ser2. Therefore FP treatment creates a confounding factor in these ChIP experiments that the reviewer feels nullifies any major conclusions that could be made. Additionally, the authors provide no validation data for the siRNA to show knockdown used in these same experiments, and no replicate number is included. The wide standard deviations and lack of an indication of significance in 5e lead the reviewer to believe low replicate number. Finally, any details on how the Pausing Ratio is calculated for these data are not included in this manuscript.

Response:

- (1) While the reviewer correctly points out that FP is an inhibitor of PTEFb/Cdk9 kinase responsible for phosphorylating Pol II on Ser2, we respectfully disagree with the conclusion that flavopiridol (FP) treatment confounds the use of an antibody to track Ser 2 phosphorylated Pol II traveling through the genes. Similar timed ChIP-seq studies in previous reports have found that FP treatment compromises the occurrence of new Pol II phosphorylation events, but it does not affect existing Ser2 phosphorylated Pol II travel within the gene bodies (Jonkers, et al. *Elife*, **3**, e02407, 2014; Jonkers and Lis, *Nature Reviews Mol Cell Biol*, 16:167-177, 2015). Thus, FP can be used to block release of new Pol II into productive elongation in a timed manner, yet Pol II traveling dynamics can be studied by detecting the “retreating wave” of Ser2 phosphorylated Pol II already in the gene bodies. Another way to study Pol II traveling is to first treat the cells with FP for 1 h. After wash-out of FP, the “emerging wave” of newly released Ser 2 Pol II can be studied to track Pol II traveling dynamics. These two complementary strategies, summarized in Rebuttal Figure 1 below (modified from Figure 3b in Jonkers and Lis, *Nature Reviews Mol Cell Biol*, 16:167-177, 2015), have been used in this report to study Ser 2 Pol II recycling (Fig. 4a-f in revised manuscript). We agree that the previous version of the manuscript did not provide a sufficient rationale, and we have added an introduction to these two types of timed ChIP-seq analyses in the paragraph “PAF1 is required for RNA Pol II transcriptional recycling *in vivo*” on manuscript page 10.

Rebuttal Figure 1. Schematic for timed ChIP-seq analysis used for studying Pol II traveling dynamics. (A) Timed treatment of FP to block release of new Pol II into gene bodies. **(B)** FP treatment followed by wash-out of FP and timed release of new Pol II into gene bodies.

- (2) Validation data for siPAF1 has been provided in **New Supplementary Fig 6a** and replicate numbers of ChIP-seq (**n=5**) have been included in GEO under accession number GSE133655.
- (3) We would like to clarify that the wide standard deviations in the previous Figure 5e (Figure 4e in the revised manuscript) were caused by biological variability among the 5,000 genes included in the analysis and not by low replicate number. The indication of significance has been included in the **revised Figure 4e**. In the paragraph entitled “PAF1

is required for RNA Pol II transcriptional recycling *in vivo*” on manuscript page 11, we have added a statement that; “Ser2 phosphorylated Pol II signal in the promoter regions after 20 min of release from FP decreased 47.7% and 15.5% in the siControl group and the siPAF1 group, respectively.”

- (4) A Pausing Ratio calculation has been added to the paragraph “Sequencing data analysis” in the Methods section, on manuscript page 25. Briefly, to calculate the pause and release ratio, we first normalized all time-resolved ChIP-seq datasets of Pol II (Ser2) and PAF1 to the same sequencing depth of 50 million. Then, each transcript was divided into three regions, including a 5'-end bin [TSS -50 bp, TSS +200 bp], a gene body bin [TSS +200 bp, TTS -500 bp] and a 3'-end bin [TTS -500 bp, TTS +500 bp]. The average density per nucleotide was calculated using the normalized density files for each bin. The 5' pausing ratio compares the 5'-end density to the density in the gene body, while the 3' release ratio compares the 3'-end density to the gene body density. We calculated the ratios for Pol II (Ser2) and PAF1 occupancy at the indicated time points.

7. Figure 6. In 6C the authors again provide no quantitation of Paf1 depletions, and visually/qualitatively it looks modest at best. 6a/b does seem to show elevated Paf1 levels in prostate tumor samples. However, this conclusion creates a confounding factor for any experiments where the prostate cancer cell line (LNCaP-abl) was used, as it seems likely this cell line could also then have high Paf1 levels. However, this possible limitation was not discussed or explored.

Response:

- 1) Quantification has been provided for PAF1 depletions (**New Fig. 5c lower panel, and New Supplementary Fig. 7a**).
- 2) To address the concern about likely PAF1 overexpression in the LNCaP-abl cancer cell model, we also performed the transcriptional recycling assay using nuclear extract from non-cancerous BPH1 cells, a human benign prostatic hyperplasia cell line that expresses lower PAF1 than LNCaP-abl (**New Supplementary Fig. 5c**). While transcriptional activity was more than 3-fold lower in transcription recycling assays using nuclear extracts from BPH1 as opposed to LNCaP-abl, moderate depletion of PAF1 from BPH1 nuclear extract still significantly decreased transcriptional recycling, although to a lesser extent (**New Supplementary Fig. 5d**).

8. Altogether, the data doesn't seem strong enough or direct enough to conclude that Paf1 is a driver of recycling. It does seem likely that Paf1 is required for full efficiency of transcription elongation and that depleting it leaves RNAPII more open to alternative termination mechanisms before reaching the canonical termination site. However, this manuscript does not seem to consider that possibility, or any other data-based alternatives when presenting the model of Paf1 as a driver of recycling. Overall, the lack of rigor and reproducibility in these experiments, still leaves the reviewer unable to agree with any significant conclusions made from these data.

Response:

- (1) We hope that these concerns about the strength of the studies have been addressed by the additional experiments discussed in the above Responses to Specific points 1-7. We have

followed the reviewer's suggestions to perform all required experiments in at least triplicate with rigorous experimental design, and have conducted rigorous evaluation of previous studies. We believe that the additional experiments have added substantially to the merit of the manuscript, and find that they combine with our integration of *in vitro* systems with studies in cells and clinical samples to strongly and consistently support the conclusion that PAF1 is a driver for transcriptional recycling.

- (2) We appreciate the reviewer's interesting point about alternative transcriptional termination mechanisms before reaching the canonical termination site following PAF1 knockdown. Indeed, we have come across a previous study showing that depletion of PAF1 may promote noncanonical TSS proximal polyadenylation sites (PAS) usage in mouse myoblasts, which may be due to increased Pol II binding at the TSS proximal regions after PAF1 knockdown and subsequent stimulation of the TSS proximal PAS usage (Yang et al. *PloS Genet*, 12: e1005794, 2016). We analyzed our timed Ser 2 Pol II ChIP-seq data in the presence or absence of PAF1 during 0-, 10-, or 20- minute recovery time points after wash-out of FP to release Pol II into the gene bodies (see Rebuttal Figure 1B). Surprisingly, our results showed that Pol II levels were decreased in PAF1- versus control-silenced cells in TSS proximal regions (**New Supplementary Fig. 8a**), suggesting that PAF1 might not promote noncanonical TSS proximal PAS usage in our human cell model. Nevertheless, this raises an interesting subject for future investigation of mechanism.
- (3) We have also examined the effect of PAF1 knockdown on transcription elongation rates (please see Response to Reviewer 1 point 6). Using our Pol II Ser 2 timed ChIP-seq data generated during 10-, 30-, or 40-minute FP treatment to block release of new Pol II into gene bodies (see Rebuttal Figure 1A), we found that the presence of PAF1 can accelerate elongation rates (**New Supplementary Fig. 8b, c**). This result is consistent with recent findings that loss of Paf1 results in Pol II elongation rate defects with significant rate compression (Hou et al. *PNAS*, 116: 14583-14592, 2019).

In summary, our evidence-based studies suggest that PAF1 can drive transcriptional recycling and accelerate elongation rates. These abilities are consistent with the emerging notion that PAF1 can regulate many aspects of Pol II transcription and that our understanding of the molecular functions of PAF1 is continually expanding (Van Oss et al., *Trends Biochem Sci* 42, 788-798, 2017). We have revised our discussion of the role of PAF1 in transcription regulation within the "Discussion" section (pages 15-16).

Minor points

1. *Supplementary Figure 4 is used to conclude that Leo1 and WRD61 depletion do not exacerbate the effect of Paf1 depletion. The authors did not use proper control of individually depleting Leo1 or WRD61.*

Response: The reviewer suggestion proved to be an important point. We have now depleted WDR61 or LEO1 individually, in order to determine the role of each PAF1 complex subunit in transcription recycling. Depletion of PAF1 or LEO1 but not WDR61 caused depletion of other PAF complex subunits and led to decreased transcription recycling (**New Supplementary Fig. 5f, g**). Our results indicate that a PAF1 complex containing structurally and genomically tightly associated PAF1 and LEO1 (Chu et al. *Nucleic Acids Res*, 41: 10619-10629, 2013 and **New Supplementary Fig. 4g-i**) is important to enable Pol II transcription to recycle, while WDR61 is

dispensable for transcription recycling. We have revised our conclusion accordingly in the paragraph within the Results section entitled “PAF1 depletion attenuates RNA Pol II transcriptional recycling *in vitro*” (Page 10).

Reviewer #3 (Remarks to the Author):

This manuscript presents a novel methodological approach for assessing different stages of gene regulation. Importantly, this permits the analysis and comparison between expression from the first template, versus expression from subsequent templates. The authors show that the protein machinery from the first template was capable of regulating expression from the second template, which implies that RNA Pol2 is recycled. This method uses an in vitro template coupled with cell lysates, permitting the discovery of known and unknown components of the process. The authors couple this capture approach with 2D gels and Mass Spec to identify proteins involved in the different stages of gene expression, revealing a number of known factors and confirming that discrete complexes exist. By exploring proteins that are enriched in the recycling phase, the authors identify the PAF1 complex as being significantly biased towards the second template, suggesting that the PAF1 complex is involved in the recycling process. A series of ChIP-seq experiments are undertaken to show that PAF1 is recycled to the promoters of target genes in a prostate cancer cell line model. Data is provided to show that depletion of PAF1 doesn't impact expression from the first template, but has profound consequences on expression from the second template, confirming a functional role for the PAF1 complex in recycling. This is further validated by assessing RNA Pol2 recycling in growing cells that have been depleted for PAF1. Finally, PAF1 is shown to be overexpressed in prostate cancer relative to adjacent normal or healthy tissue.

This is a very good manuscript. The method is novel and is exploited in a smart and effective manner, revealing new insight into the fundamentals of gene regulation and complex recycling. The paper is well written and the data/figures are clear and convincing. I have a few issues that need to be addressed, but I think this would be an important contribution to the scientific community.

Response: We thank the reviewer for appreciating the novelty and impact of our study.

- It wasn't clear (even after wading through the methods section) how many replicates were conducted of the Mass Spec discovery, how replicates were integrated or what the degree of variability is. Can the authors do a sample swap (i.e. purposely match up samples from different experimental conditions as 'replicates') to identify the degree of variability in the system? Given this is a new method, some confidence in the number of proteins that are 'differential' by chance would be an important piece of information.

Response: We have conducted new biological triplicates with technical triplicates for each condition, and generated a total of 36 new samples for LC-MS/MS. As shown in **New Supplementary Fig. 2a-c**, the sample correlation coefficient among both biological triplicates and technical triplicates is 0.95~0.97. We have conducted sample swap as suggested by the reviewer, and performed unsupervised clustering for all samples. The results indicate that the proteins differentially enriched within different transcription steps are consistent only among

both biological triplicates and technical triplicates (**New Supplementary Fig. 2d**). On the other hand, purposefully matching up samples from different conditions as ‘replicates’ failed to reproduce results (**New Supplementary Fig. 2e, f**), demonstrating that the differentially enriched proteins we identified are convincingly reproducible and not identified by chance.

- It wasn't clear why the authors did the initial discovery in HeLa cells but then jumped into the LNCaP-abl cells? What was the logic behind this and based on the final IHC data, is this the right model to be making conclusions about PAF1 function, given the inference that PAF1 levels will be overexpressed and not representative of physiologic levels (or potentially function).

Response: HeLa nuclear extract is widely used for *in vitro* transcription assays because it has been established as a source of transcription-related factors and proteins that support accurate transcription initiation by Pol II, and displays both basal and regulated transcription (Dignam et al. *Nucleic Acids Res.* 11, 1475–89, 1983; Sawdogo and Roeder. *PNAS*, 82:4394-4398, 1985). While we chose to use HeLa nuclear extracts in our *in vitro* transcription system for these reasons, we then changed to LNCaP-abl prostate cancer cells as an intact cell model in which to characterize *in vitro* identified transcription recycling factors and to distinguish “run-off” and “drop-off” factors. We see the reviewer’s point about the caveats of using a cell line in which PAF1 is likely to be overexpressed. To ease the concern about the LNCaP-abl cancer cell model, we have now repeated the transcriptional recycling assay using nuclear extract from non-cancerous BPH1 cells, a human benign prostatic hyperplasia cell line that expresses PAF1 at a lower level than LNCaP-abl (**New Supplementary Fig. 5c**). While transcriptional activity is more than 3-fold lower in transcription recycling assays using nuclear extracts from BPH1 as opposed to LNCaP-abl, moderate depletion of PAF1 from BPH1 nuclear extract still significantly decreased transcriptional recycling, although to a lesser extent (**New Supplementary Fig. 5d**).

- To this point, what is the relevance of overexpressed PAF1 in cancer versus normal and will this impact the efficiency of the second template activity that might only be observed in cancer models and not representative of normal gene expression properties in non-cancerous contexts?

Response: This is an interesting question that we hope has been satisfactorily addressed by the experiment described in the previous response above. Briefly, to address this concern raised by the reviewer, we have now examined the role of PAF1 in transcription recycling using a non-cancerous prostate cell line, BPH1. Although BPH1 expressed lower level of PAF1 compared to LNCaP-abl, PAF1 loss in BPH1 still significantly decreased transcription recycling in non-cancerous contexts, though to a less extent (**New Supplementary Fig. 5c, d**) (see also Response to the above question).

- Is there any published ChIP-seq data for PAF1, LEO1 and WDR61? Do they always co-bind or is there a potential role for these proteins independently of the other two proteins?

Response: To address the question of whether subunits of the PAF1 complex co-bind across the genome, we have performed LEO1 and RTF1 ChIP-seq within the same LNCaP-abl cell line we used to generate PAF1 ChIP-seq data. Please note that we did not perform WDR61 ChIP-seq, as other new data had already found that WDR61 depletion does not affect transcription recycling,

(**New Supplementary Fig. 5g**). RTF1 was selected as a replacement. As shown in **New Supplementary Fig. 4g-i**, PAF1 and LEO1 co-bound tightly throughout transcribed genes, while RTF1 co-occupied with PAF1 at TSS and gene bodies but dissociated around the transcription termination site (TTS) region. This result regarding distinct RTF1 binding patterns is consistent with the established notion that RTF1 is not strongly associated with the PAF1 complex and can function independently of other subunits in some contexts (Chu et al. *Nucleic Acids Res*, 41: 10619-10629, 2013; Van Oss et al. *Trends Biochem Sci*, 42: 788-798, 2017).

- *Given the evidence that PAF1 siRNA dramatically affects cell viability, how do the authors get confidence that the changes in RNA Pol2 recycling are because of changes in recruitment to promoters, as opposed to indirect consequences of decreases in general cell viability and health?*

Response: We agree with the reviewer that RNA Pol II recycling could be caused by indirect consequences of cell viability decrease triggered by PAF1 silencing. To avoid / minimize this happening, we have already used a low concentration (5 nM) of PAF siRNA in our timed ChIP-seq analyses (please see Fig. 4 in revised manuscript). The revised version of the manuscript also provides new western blot and cell growth data to demonstrate that low concentrations of PAF1 siRNA treatment moderately decrease PAF1 expression without affecting cell viability (**New Supplementary Fig. 6a**).

Minor points

- *The authors claim that the ChIP-seq measures ‘in vivo’ activity. I suspect that people who study cancer biology in animal models might take exception to this.*

Response: We see the reviewer’s point and have used “in human cells” or “cellular” to replace the term “in vivo”.

- *The authors claim that PAF1 could be a relevant drug target because ‘PAF1 as a potentially favorable therapeutic target due to its categorization as a context-specific cell fitness gene that is unlikely to perform essential functions in normal cells and may therefore be associated with less side effects’. I think it’s premature to make this statement. Analysis of the Broad/Novartis DepMap suggests that 715 out of 726 cancer cell lines are dependent on PAF1. So, either PAF1 is required in almost all cancer cells, but in no normal tissue, which would require a systematic analysis of PAF1 dependence in a panel of normal cell models. Or more likely, PAF1 is required in all models, regardless of tissue of origin, whether the model is normal or diseased etc, in which case, PAF1 is unlikely to be a bone fide therapeutic target.*

Response: We agree with the reviewer and have removed the statement that “PAF1 as a potentially favorable therapeutic target due to its categorization as a context-specific cell fitness gene that is unlikely to perform essential functions in normal cells and may therefore be associated with less side effects”.

Reviewers' Comments:

Reviewer #1:

Remarks to the Author:

This reviewer is satisfied with the rebuttal responses to my previous criticisms.

Reviewer #3:

Remarks to the Author:

The authors have addressed all of my concerns. The inclusion of the new data directly addresses my criticisms and the justification of the cell lines used is valid.